# LEARNING LATENT STRUCTURAL CAUSAL MODELS

## ABSTRACT

Causal learning has long concerned itself with the accurate recovery of underlying causal mechanisms. Such causal modelling enables better explanations of out-of-distribution data. Prior works on causal learning assume that the high-level causal variables are given. However, in machine learning tasks, one often operates on low-level data like image pixels or high-dimensional vectors. In such settings, the entire Structural Causal Model (SCM) – structure, parameters, *and* high-level causal variables – is unobserved and needs to be learnt from low-level data. We treat this problem as Bayesian inference of the latent SCM, given low-level data. For linear Gaussian additive noise SCMs, we present a tractable approximate inference method which performs joint inference over the causal variables, structure and parameters of the latent SCM from random, known interventions. Experiments are performed on synthetic datasets and a causally generated image dataset to demonstrate the efficacy of our approach. We also perform image generation from unseen interventions, thereby verifying out of distribution generalization for the proposed causal model.

## 1 INTRODUCTION

Learning variables of interest and uncovering causal dependencies is crucial for intelligent systems to reason and predict in scenarios that differ from the training distribution. In the causality literature, causal variables and mechanisms are often assumed to be known. This knowledge enables reasoning and prediction under unseen interventions. In machine learning, however, one does not have direct access to the underlying variables of interest nor the causal structure and mechanisms corresponding to them. Rather, these have to be learned from observed low-level data like pixels of an image which are usually high-dimensional. Having a learned causal model can then be useful for generalizing to out-of-distribution data (Scherrer et al., 2022; Ke et al., 2021), estimating the effect of interventions (Pearl, 2009; Schölkopf et al., 2021), disentangling underlying factors of variation (Bengio et al., 2012; Wang and Jordan, 2021), and transfer learning (Schoelkopf et al., 2012; Bengio et al., 2019).

Structure learning (Spirtes et al., 2000; Zheng et al., 2018) learns the structure and parameters of the Structural Causal Model (SCM) (Pearl, 2009) that best explains some observed high-level causal variables. In causal machine learning and representation learning, however, these causal variables may no longer be observable. This serves as the motivation for our work. We address the problem of learning the entire SCM – consisting its causal variables, structure and parameters – which is latent, by learning to generate observed low-level data. Since one often operates in low-data regimes or non-identifiable settings, we adopt a Bayesian formulation so as to quantify epistemic uncertainty over the learned latent SCM. Given a dataset, we use variational inference to learn a joint posterior over the causal variables, structure and parameters of the latent SCM. To the best of our knowledge, ours is the first work to address the problem of Bayesian causal discovery in linear Gaussian latent SCMs from low-level data, where causal variables are unobserved. Our contributions are as follows:

- We *propose a general algorithm for Bayesian causal discovery in the latent space of a generative model*, learning a distribution over causal variables, structure and parameters in linear Gaussian latent SCMs with random, known interventions. Figure 1 illustrates an overview of the proposed method.

- By learning the structure and parameters of a latent SCM, we implicitly induce a joint distribution over the causal variables. Hence, sampling from this distribution is equivalent to ancestral sampling through the latent SCM. As such, *we address a challenging, simultane-*

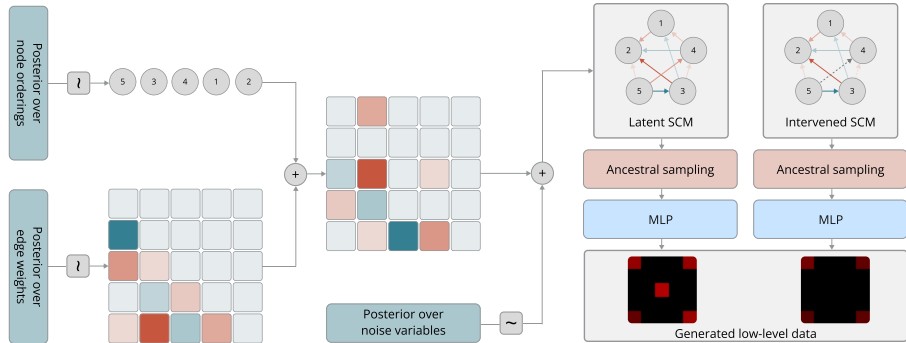

Figure 1: Model architecture of the proposed generative model for the Bayesian latent causal discovery task to learn latent SCM from low-level data.

*ous optimization problem* that is often encountered during causal discovery in latent space: one cannot find the right graph without the right causal variables, and vice versa.

- On a synthetically generated dataset and an image dataset used to benchmark causal model performance (Ke et al., 2021), we evaluate our method along three axes – uncovering causal variables, structure, and parameters – consistently outperforming baselines. We demonstrate its ability to perform image generation from unseen interventional distributions.

## 2 PRELIMINARIES

### 2.1 STRUCTURAL CAUSAL MODELS

A Structural Causal Model (SCM) is defined by a set of equations which represent the mechanisms by which each endogenous variable $Z_i$ depends on its direct causes $Z^{\mathcal{G}}_{pa(i)}$ and a corresponding exogenous noise variable $\epsilon_i$. The direct causes are subsets of other endogenous variables. If the causal parent assignment is assumed to be acyclic, then an SCM is associated with a Directed Acyclic Graph (DAG) $\mathcal{G} = (V, E)$, where V corresponds to the endogenous variables and $E$ encodes direct cause-effect relationships. The exact value $z_i$ taken on by a causal variable $Z_i$, is given by local causal mechanisms $f_i$ conditional on the values of its parents $z^{\mathcal{G}}_{pa(i)}$, the parameters $\Theta_i$, and the node's noise variable $\epsilon_i$, as given in equation 1.

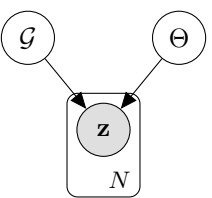

Figure 2: BN for prior works in causal discovery and structure learning

For linear Gaussian additive noise SCMs with equal noise variance, i.e., the setting that we focus on in this work, all $f_i$'s are linear functions, and $\Theta$ denotes the weighted adjacency matrix $W$, where each $W_{ji}$ is the edge weight from $j \rightarrow i$. The linear Gaussian additive noise SCM thus reduces to equation 2,

$$z_i = f_i(z^{\mathcal{G}}_{pa(i)}, \Theta, \epsilon_i), \quad (1) \qquad z_i = \sum_{j \in pa_{\mathcal{G}}(i)} W_{ji} \cdot z_j + \epsilon_i. \quad (2)$$

### 2.2 CAUSAL DISCOVERY

Structure learning in prior work refers to learning a DAG according to some optimization criterion with or without the notion of causality (e.g., He et al. (2019)). The task of causal discovery on the other hand, is more specific in that it refers to learning the structure (also parameters, in some cases) of SCMs, and subscribes to causality and interventions like that of Pearl (2009). That is, the methods aim to estimate $(\mathcal{G}, \Theta)$. These approaches often resort to modular likelihood scores over causal variables – like the BGe score (Geiger and Heckerman, 1994; Kuipers et al., 2022) and BDe

score (Heckerman et al., 1995) – to learn the right structure. However, these methods all assume a dataset of observed causal variables. These approaches either obtain a maximum likelihood estimate,

$$\mathcal{G}^* = \arg\max_{\mathcal{G}} p(Z \mid \mathcal{G}) \quad \text{or} \quad (\mathcal{G}^*, \Theta^*) = \arg\max_{\mathcal{G}, \Theta} p(Z \mid \mathcal{G}, \Theta), \tag{3}$$

or in the case of Bayesian causal discovery (Heckerman et al., 1997), variational inference is typically used to approximate a joint posterior distribution $q_\phi(\mathcal{G}, \Theta)$ to the true posterior $p(\mathcal{G}, \Theta \mid Z)$ by minimizing the KL divergence between the two,

$$D_{\text{KL}}(q_\phi(\mathcal{G}, \Theta) \,\|\, p(\mathcal{G}, \Theta \mid Z)) = -\mathbb{E}_{(\mathcal{G}, \Theta) \sim q_\phi}\left[\log p(Z \mid \mathcal{G}, \Theta) - \log \frac{q_\phi(\mathcal{G}, \Theta)}{p(\mathcal{G}, \Theta)}\right], \tag{4}$$

where $p(\mathcal{G}, \Theta)$ is a prior over the structure and parameters of the SCM – possibly encoding DAGness. Figure 2 shows the Bayesian Network (BN) over which inference is performed for causal discovery tasks.

### 2.3 LATENT CAUSAL DISCOVERY

In more realistic scenarios, the learner does not directly observe causal variables and they must be learned from low-level data. The causal variables, structure, and parameters are part of a latent SCM. The goal of causal representation learning models is to perform inference of, and generation from, the true latent SCM. Yang et al. (2021) proposes a Causal VAE but is in a supervised setup where one has labels on causal variables and the focus is on disentanglement. Kocaoglu et al. (2018) present causal generative models trained in an adversarial manner but assumes observations of causal variables. Given the right causal structure as a prior, the work focuses on generation from conditional and interventional distributions.

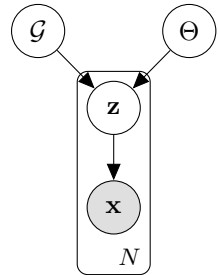

Figure 3: BN for the latent causal discovery task that generalizes standard causal discovery setups

In both the causal representation learning and causal generative model scenarios mentioned above, the Ground Truth (GT) causal graph and parameters of the latent SCM are arbitrarily defined on real datasets and the setting is supervised. Contrary to this, our setting is unsupervised and we are interested in recovering the GT underlying SCM and causal variables that generate the low-level observed data – we define this as the problem of *latent causal discovery*, and the BN over which we want to perform inference on is given in figure 3. In the upcoming sections, we discuss related work, formulate our problem setup and propose an algorithm for Bayesian latent causal discovery, evaluate with experiments on causally created vector data and image data, and perform sampling from unseen interventional image distributions to showcase generalization of learned latent SCMs.

## 3 RELATED WORK

Prior work can be classified into Bayesian (Koivisto and Sood, 2004; Heckerman et al., 2006; Friedman and Koller, 2013) or maximum likelihood (Brouillard et al., 2020; Wei et al., 2020; Ng et al., 2022) methods, that learn the structure and parameters of SCMs using either score-based (Kass and Raftery, 1995; Barron et al., 1998; Heckerman et al., 1995) or constraint-based (Cheng et al., 2002; Lehmann and Romano, 2005) approaches.

**Causal discovery:** Work in this category assume causal variables are observed and do not operate on low-level data (Spirtes et al., 2000; Viinikka et al., 2020; Yu et al., 2021; Zhang et al., 2022). Peters and Bühlmann (2014) prove identifiability of linear Gaussian SCMs with equal noise variances. Bengio et al. (2019) use the speed of adaptation as a signal to learn the causal direction. Ke et al. (2019) explore learning causal models from unknown, while Scherrer et al. (2021); Tigas et al. (2022); Agrawal et al. (2019); Toth et al. (2022) focus on active learning and experimental design setups on how to perform interventions to efficiently learn causal models. Transformer (Vaswani et al., 2017) based approach learns structure from synthetic datasets and generalize to naturalistic graphs (Ke et al., 2022). Zheng et al. (2018) introduce an acyclicity constraint that penalizes cyclic

graphs, thereby restricting search close to the DAG space. Lachapelle et al. (2019) leverages this constraint to learn DAGs in nonlinear SCMs. Pamfil et al. (2020); Lippe et al. (2022) perform structure learning on temporal data.

**Structure learning with latent variables**: Markham and Grosse-Wentrup (2020) introduces the concept of Measurement Dependence Inducing Latent Causal Models (MCM). The proposed algorithm finds a minimal-MCM that induces the dependencies between observed variables. However, similar to VAEs, the method assumes no causal links between latent variables. Kivva et al. (2021) provides the conditions under which the number of latent variables and structure can be uniquely identified for discrete latent variables, given the adjacency matrix between the hidden and measurement variables has linearly independent columns. Elidan et al. (2000) detects the signature of hidden variables using semi-cliques and then performs structure learning using the structural-EM algorithm (Friedman, 1998) for discrete random variables. Anandkumar et al. (2012) and Silva et al. (2006) considers the identifiability of linear Bayesian Networks when some variables are unobserved. In the former work, the identifiability results hold only for particular classes of DAGs which follow certain structural constraints. Assuming non-Gaussian noise and that certain sets of latents have a lower bound on the number of pure measurement child variables, Xie et al. (2020) proposes the GIN condition to identify the structure between latent confounders. The formulation in the above works involves SCMs where some variables are observed and the others are unobserved. In contrast, the entire SCM is latent in our setup. Finally, GraphVAE (He et al., 2019) learns a structure between latent variables but does not incorporate notions of causality.

**Supervised causal representation learning**: Brehmer et al. (2022) present identifiability theory for learning causal representations and propose an algorithm assuming access to pairs of observational and interventional data. Ahuja et al. (2022a) studies identifiability in a similar setup with the use of sparse perturbations. Ahuja et al. (2022c) discusses identifiability for causal representation learning when one has access to interventional data. Kocaoglu et al. (2018); Shen et al. (2021); Moraffah et al. (2020) introduce generative models that use an SCM-based prior in latent space. In Shen et al. (2021), the goal is to learn causally disentangled variables. Yang et al. (2021) learn a DAG but assumes complete access to the causal variables. Lopez-Paz et al. (2016) establishes observable causal footprints in images. We refer the reader to section A.9 for a discussion on identifiability.

## 4 LEARNING LATENT SCMs FROM LOW-LEVEL DATA

### 4.1 PROBLEM SCENARIO

We are given a dataset $\mathcal{D} = \{\mathbf{x}^{(1)}, ..., \mathbf{x}^{(N)}\}$, where each $\mathbf{x}^{(i)}$ is a high-dimensional observed data – for simplicity, we assume $\mathbf{x}^{(i)}$ is a vector in $\mathbb{R}^D$ but the setup extends to other inputs as well. We assume that there exist latent causal variables $\mathbf{Z} = \{\mathbf{z}^{(i)} \in \mathbb{R}^d\}_{i=1}^N$ where $d \leq D$, that explain the data $\mathcal{D}$, and these latent variables belong to a GT SCM $\mathcal{G}_{GT}, \Theta_{GT}$. We wish to invert the data generation process $g : (\mathbf{Z}, \mathcal{G}_{GT}, \Theta_{GT}) \rightarrow \mathcal{D}$. In the setting, we also have access to the intervention targets $\mathcal{I} = \{\mathcal{I}^{(i)}\}_{i=1}^N$ where each $\mathcal{I}^{(i)} \in \{0,1\}^d$. The $j^{\text{th}}$ dimension of $\mathcal{I}^{(i)}$ takes a value of 1 if node $j$ was intervened on in data sample $i$, and 0 otherwise. We will take $\mathcal{X}, \mathcal{Z}, \mathcal{G}, \Theta$ to be random variables over low-level data, latent causal variables, the SCM structure, and SCM parameters respectively.

### 4.2 GENERAL METHOD

We aim to obtain a posterior estimate over the entire latent SCM, $p(\mathcal{Z}, \mathcal{G}, \Theta \mid \mathcal{D})$. Computing the true posterior analytically requires calculating the marginal likelihood $p(\mathcal{D})$ which gets quickly intractable due to the number of possible DAGs growing super-exponentially with respect to the number of nodes. Thus, we resort to variational inference (Blei et al., 2017) that provides a tractable way to learn an approximate posterior $q_\phi(\mathcal{Z}, \mathcal{G}, \Theta)$ with variational parameters $\phi$, close to the true posterior $p(\mathcal{Z}, \mathcal{G}, \Theta \mid \mathcal{D})$ by maximizing the Evidence Lower Bound (ELBO),

$$\mathcal{L}(\psi, \phi) = \mathop{\mathbb{E}}_{q_\phi(\mathcal{Z}, \mathcal{G}, \Theta)} \left[ \log p_\psi(\mathcal{D} \mid \mathcal{Z}, \mathcal{G}, \Theta) - \log \frac{q_\phi(\mathcal{Z}, \mathcal{G}, \Theta)}{p(\mathcal{Z}, \mathcal{G}, \Theta)} \right], \qquad (5)$$

where $p(\mathcal{Z}, \mathcal{G}, \Theta)$ is the prior, $p_\psi(\mathcal{D} \mid \mathcal{Z}, \mathcal{G}, \Theta)$ is the likelihood model with parameters $\psi$, the likelihood model maps the latent variables to high-dimensional vectors. An approach to learn this

posterior could be to factorize it as

$$q_\phi(\mathcal{Z}, \mathcal{G}, \Theta) = q_\phi(\mathcal{Z}) \cdot q_\phi(\mathcal{G}, \Theta \mid \mathcal{Z}) \tag{6}$$

Given a way to obtain $q_\phi(\mathcal{Z})$, the conditional $q_\phi(\mathcal{G}, \Theta \mid \mathcal{Z})$ can be obtained using existing Bayesian structure learning methods. Otherwise, one has to perform a hard simultaneous optimization which would require alternating optimizations on $\mathcal{Z}$ and on $(\mathcal{G}, \Theta)$ given an estimate of $\mathcal{Z}$. Difficulty of such an alternate optimization is discussed in Brehmer et al. (2022).

**Alternate factorization of the posterior**: Rather than factorizing as in equation 6, we propose to only introduce a variational distribution $q_\phi(\mathcal{G}, \Theta)$ over structures and parameters, so that the approximation is given by $q_\phi(\mathcal{Z}, \mathcal{G}, \Theta) = p(\mathcal{Z} \mid \mathcal{G}, \Theta) \cdot q_\phi(\mathcal{G}, \Theta)$. The advantage of this factorization is that the true distribution $p(\mathcal{Z} \mid \mathcal{G}, \Theta)$ over $\mathcal{Z}$ is completely determined from the SCM given $(\mathcal{G}, \Theta)$ and exogenous noise variables (assumed to be Gaussian). This conveniently avoids the hard simultaneous optimization problem mentioned above since optimizing for $q_\phi(\mathcal{Z})$ is not necessary. Hence, equation 5 simplifies to:

$$\mathcal{L}(\psi, \phi) = \underset{q_\phi(\mathcal{Z}, \mathcal{G}, \Theta)}{\mathbb{E}} \left[ \log p_\psi(\mathcal{D} \mid \mathcal{Z}) - \log \frac{q_\phi(\mathcal{G}, \Theta)}{p(\mathcal{G}, \Theta)} - \log \frac{p(\mathcal{Z} \mid \mathcal{G}, \Theta)}{p(\mathcal{Z} \mid \mathcal{G}, \Theta)}^{\,0} \right] \tag{7}$$

Such a posterior can be used to obtain an SCM by sampling $\hat{\mathcal{G}}$ and $\hat{\Theta}$ from the approximated posterior. As long as the samples $\hat{\mathcal{G}}$ are always acyclic, one can perform ancestral sampling through the SCM to obtain predictions of the causal variables $\hat{\mathbf{z}}^{(i)}$. For additive noise models like in equation 2, these samples are already reparameterized and differentiable with respect to their parameters. The samples of causal variables are then fed to the likelihood model to predict samples $\hat{\mathbf{x}}^{(i)}$ that reconstruct the observed data $\mathbf{x}^{(i)}$.

### 4.3   POSTERIOR PARAMETERIZATIONS AND PRIORS

For linear Gaussian latent SCMs, which is the focus of this work, learning a posterior over $(\mathcal{G}, \Theta)$ is equivalent to learning $q_\phi(W, \Sigma)$ – a posterior over weighted adjacency matrices $W$ and noise covariances $\Sigma$. We follow an approach similar to (Cundy et al., 2021). We express $W$ via a permutation matrix $P$[1] and a lower triangular edge weight matrix $L$, according to $W = P^T L^T P$. Here, $L$ is defined in the space of all weighted adjacency matrices with a fixed node ordering where node $j$ can be a parent of node $i$ only if $j > i$. Search over permutations corresponds to search over different node orderings and thus, $W$ and $\Sigma$ parameterize the space of SCMs. Further, we factorize the approximate posterior $q_\phi(P, L, \Sigma)$ as

$$q_\phi(\mathcal{G}, \Theta) \equiv q_\phi(W, \Sigma) \equiv q_\phi(P, L, \Sigma) = q_\phi(P \mid L, \Sigma) \cdot q_\phi(L, \Sigma) \tag{8}$$

Combining equation 7 and 8 leads to the following ELBO which has to be maximized (derived in A.1), and the overall method is summarized in algorithm 1,

$$\mathcal{L}(\psi, \phi) = \underset{q_\phi(L, \Sigma)}{\mathbb{E}} \left[ \underset{q_\phi(P \mid L, \Sigma)}{\mathbb{E}} \left[ \underset{q_\phi(\mathcal{Z} \mid P, L, \Sigma)}{\mathbb{E}} \left[ \log p_\psi(\mathcal{D} \mid \mathcal{Z}) \right] - \log \frac{q_\phi(P \mid L, \Sigma)}{p(P)} \right] - \log \frac{q_\phi(L, \Sigma)}{p(L) p(\Sigma)} \right] \tag{9}$$

**Distribution over** $(L, \Sigma)$: The posterior distribution $q_\phi(L, \Sigma)$ has $(\frac{d(d-1)}{2} + 1)$ elements to be learnt in the equal noise variance setting. This is parameterized as a diagonal covariance normal distribution. For the prior $p(L)$ over the edge weights, we promote sparse DAGs by using a horseshoe prior (Carvalho et al., 2009), similar to Cundy et al. (2021). A Gaussian prior is defined over $\log \Sigma$.

**Distribution over** $P$: Since the values of $P$ are discrete, performing a discrete optimization is combinatorial and becomes quickly intractable with increasing $d$. This can be handled by relaxing the discrete permutation learning problem to a continuous optimization problem. This is commonly done by introducing a Gumbel-Sinkhorn (Mena et al., 2018) distribution and where one has to calculate $S((T + \gamma)/\tau)$, where $T$ is the parameter of the Gumbel-Sinkhorn, $\gamma$ is a matrix of standard Gumbel noise, and $\tau$ is a fixed temperature parameter. The logits $T$ are predicted by passing the

---

[1] A permutation matrix $P \in \{0, 1\}^{d \times d}$ is a bistochastic matrix with $\sum_i p_{ij} = 1 \forall j$ and $\sum_j p_{ij} = 1 \forall i$.

---

**Algorithm 1** Bayesian latent causal discovery to learn $\mathcal{G}, \Theta, \mathcal{Z}$ from high dimensional data

---

**Input:** $\mathcal{D}, \mathcal{I}$
**Output:** Posterior samples over $\mathcal{G}, \Theta, \mathcal{Z}$
  1: Initialize $q_\phi(L, \Sigma), \text{MLP}_{\phi(T)}, p_\psi(\mathcal{X} \mid \mathcal{Z}), \tau$ and set learning rate $\alpha$
  2: **for** num_epochs **do**
  3:     $(\widehat{L}, \widehat{\Sigma}) \sim q_\phi(L, \Sigma)$
  4:     $T \leftarrow \text{MLP}_{\phi(T)}(\widehat{L}, \widehat{\Sigma})$             $\triangleright$ Compute logits for sampling from $q_\phi(P \mid L, \Sigma)$
  5:     $\gamma \in \mathbb{R}^{d \times d} \sim \text{standard Gumbel}$
  6:     $\widehat{P}_{\text{soft}} \leftarrow Sinkhorn((T + \gamma)/\tau)$
  7:     $\widehat{P}_{\text{hard}} \leftarrow Hungarian(\widehat{P}_{\text{soft}}; \tau \to 0)$
  8:     $\widehat{W} \leftarrow \widehat{P}^T \widehat{L}^T \widehat{P}$
  9:     **for** $i \leftarrow 1$ to $N$ **do**
 10:         $\mathcal{C}^{(i)} \leftarrow \text{argwhere}(\mathcal{I}^{(i)} = 1)$
 11:         $\widetilde{W} = \text{copy}(\widehat{W})$
 12:         $\widetilde{W}[:, \mathcal{C}^{(i)}] \leftarrow 0$         $\triangleright$ Mutated weighted adjacency matrix according to $\mathcal{I}^{(i)}$
 13:         $\widehat{W}_{\mathcal{I}^{(i)}} \leftarrow \widetilde{W}$
 14:         $\hat{\mathbf{z}}^{(i)} \leftarrow \text{AncestralSample}(\widehat{W}_{\mathcal{I}^{(i)}}, \widehat{\Sigma})$
 15:     **end for**
 16:     $\hat{\mathbf{Z}} \leftarrow \{\hat{\mathbf{z}}^{(i)}\}_{i=1}^N$
 17:     $\hat{\mathcal{D}} \sim p_\psi(\mathcal{X} \mid \mathcal{Z} = \hat{\mathbf{Z}})$
 18:     $\psi \leftarrow \psi + \alpha \cdot \nabla_\psi(\mathcal{L}(\psi, \phi))$         $\triangleright$ Update network parameters
 19:     $\phi \leftarrow \phi + \alpha \cdot \nabla_\phi(\mathcal{L}(\psi, \phi))$
 20: **end for**
 21: **return** $\text{binary}(\widehat{W}), (\widehat{W}, \widehat{\Sigma}), \hat{\mathbf{Z}}$

---

predicted $(L, \Sigma)$ through an MLP. In the limit of infinite iterations and as $\tau \to 0$, sampling from the distribution returns a doubly stochastic matrix. During the forward pass, a hard permutation $P$ is obtained by using the Hungarian algorithm (Kuhn, 1955) which allows $\tau \to 0$. During the backward pass, a soft permutation is used to calculate gradients similar to (Cundy et al., 2021; Charpentier et al., 2022). We use a uniform prior $p(P)$ over permutations.

## 5 EXPERIMENTS AND EVALUATION

We perform experiments to evaluate the learned posterior over $(\mathcal{Z}, \mathcal{G}, \Theta)$ of the true linear Gaussian latent SCM from high-dimensional data. We aim to highlight the performance of our proposed method on latent causal discovery. As proper evaluation in such a setting would require access to the GT causal graph that generated the high-dimensional observations, we test our method against baselines on synthetically generated vector data and in the realistic case of learning the SCM from pixels in the chemistry environment dataset of (Ke et al., 2021), both of which have a GT causal structure to be compared with. Further, we evaluate the ability of our model to sample images from unseen interventional distributions.

**Baselines**: Since we are, to the best of our knowledge, the first to study this setting of Bayesian learning of latent SCMs from low level observations, we are not aware of baseline methods that solve this task. However, we compare our approach against two baselines: (i) Against VAE that has a marginal independence assumption between latent variables and thus have a predefined structure in the latent space, and (ii) against GraphVAE (He et al., 2019) that learns a structure between latent variables. For all baselines, we treat the learned latent variables as causal variables and compare the recovered structure, parameters, and causal variables recovered. Since GraphVAE does not learn the parameters, we fix the edge weight over all predicted edges to be 1.

**Evaluation metrics**: *To evaluate the learned structure*, we use two metrics commonly used in the literature – the expected Structural Hamming Distance ($\mathbb{E}$-**SHD**, lower is better) obtains the SHD (number of edge flips, removals, or additions) between the predicted and GT graph and then takes an expectation over SHDs of posterior DAG samples, and the Area Under the Receiver Operating

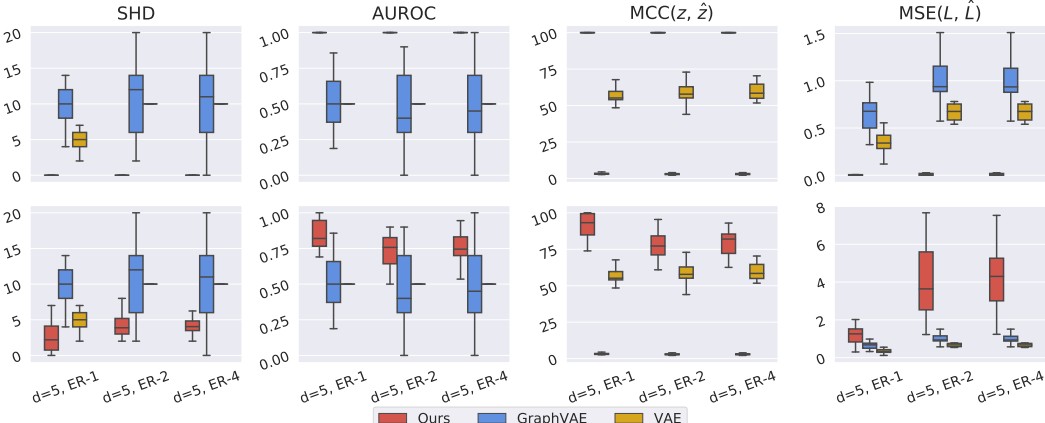

Figure 4: Learning the latent SCM (i) given a node ordering (top) and (ii) over node orderings (bottom) for **linear** projection of causal variables for $d = 5$ nodes, D $= 100$ dimensions: $\mathbb{E}$-**SHD** ($\downarrow$), **AUROC** ($\uparrow$), **MCC** ($\uparrow$), **MSE** ($\downarrow$).

Characteristic curve (**AUROC**, higher is better) where a score of 0.5 corresponds to a random DAG baseline. *To evaluate the learned parameters* of the linear Gaussian latent SCM, we use the Mean Squared Error (**MSE**, lower is better) between the true and predicted edge weights. *To evaluate the learned causal variables*, we use the Mean Correlation Coefficient (**MCC**, higher is better) following Hyvarinen and Morioka (2017); Zimmermann et al. (2021) and Ahuja et al. (2022b) which calculates a score between the true and predicted causal variables. See appendix A.3 and A.8 for training curves and more extensive evaluations of the experiments along other metrics. All our implementations are in JAX (Bradbury et al., 2018) and results are presented over 20 random DAGs.

## 5.1 Experiments on Synthetic Data

We evaluate our proposed method with the baselines on synthetically generated dataset, where we have complete control over the data generation procedure.

### 5.1.1 Synthetic Vector Data Generation

To generate high-dimensional vector data with a known causal structure, we first generate a random DAG and linear SCM parameters, and generate true causal variables by ancestral sampling. This is then used to generate corresponding high-dimensional dataset with a random projection function.

**Generating the DAG and causal variables**: Following many works in the literature, we sample random Erdős–Rényi (ER) DAGs (Erdos et al., 1960) with degrees in $\{1, 2, 4\}$ to generate the DAG. For every edge in this DAG, we sample the magnitude of edge weights uniformly as $|L| \sim \mathcal{U}(0.5, 2.0)$ and randomly sample the permutation matrix. We perform ancestral sampling through this random DAG with intervention targets $\mathcal{I}$, to obtain $\mathbf{Z}$ and then project it to D dimensions to obtain $\{\mathbf{x}^{(i)}\}_{i=1}^{N}$.

**Generating high-dimensional vectors from causal variables**: We consider two different cases of generating the high-dimensional data from the causal variables obtained in the previous step: (i) $\mathbf{x}^{(i)}$ is a random linear projection of causal variables, $\mathbf{z}^{(i)}$, from $\mathbb{R}^d$ to $\mathbb{R}^D$, according to $\mathbf{x} = \mathbf{z}\tilde{P}$, where $\tilde{P} \in \mathbb{R}^{d \times D}$ is a random projection matrix. (ii) $\mathbf{x}^{(i)}$ is a nonlinear projection of causal variables, $\mathbf{z}^{(i)}$, modeled by a 3-layer MLP.

### 5.1.2 Results on Synthetic Vector Data

**Results on linear projection of causal variables**: We present results on the learned causal variables, structure, and parameters in two scenarios: (i) when the true node ordering or permutation is given (e.g., as in He et al. (2019)), and (ii) when the node ordering is *not* given and one has to additionally also infer the permutation $P$. For $d = 5, 10, 20$ nodes projected to D $= 100$ dimensions, we evaluate our algorithm on synthetic ER-1, ER-2, and ER-4 DAGs. The model was trained for 5000

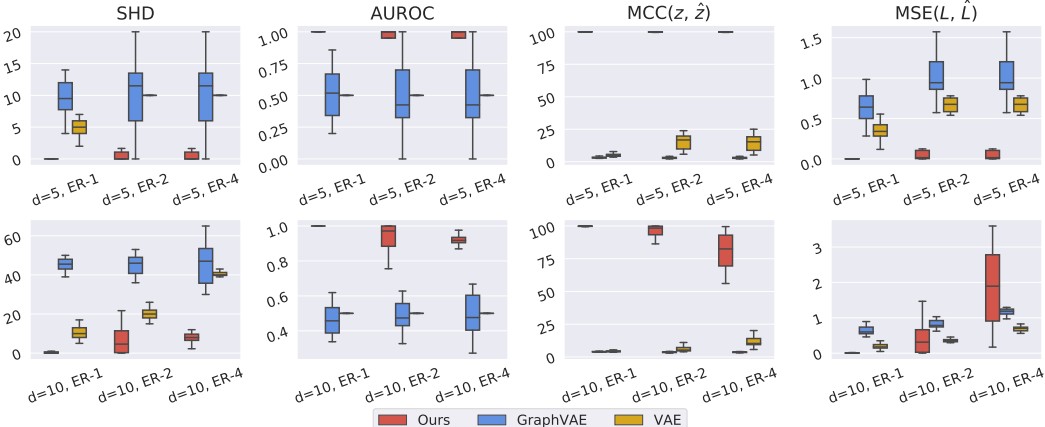

Figure 5: Learning the latent SCM for **nonlinear** projection of causal variables for $d = 5$ (top) and $d = 10$ (bottom) nodes, D = 100 dimensions, given the node ordering. $\mathbb{E}$-**SHD** ($\downarrow$), **AUROC** ($\uparrow$), **MCC** ($\uparrow$), **MSE** ($\downarrow$).

epochs so as to reach convergence. Figure 4 summarizes the results for $d = 5$ nodes, for which we use 500 observational data points and 2000 interventional data points. Of the 2000 interventional data, we generate 100 random interventional data points per set over 20 intervention sets[2]. It can be seen that when permutation is given, the proposed method can recover the causal graph correctly in all the cases, achieving $\mathbb{E}$-SHD of 0 and AUROC of 1. When the permutation is learned, the proposed method still recovers the true causal structure very well. However, this is not the case with baseline methods of VAE and GraphVAE, which perform significantly worse on most of the metrics. Figure 11 and 12 (in Appendix) show the results for $d = 10$ and $d = 20$ nodes.

**Results on nonlinear projection of causal variables**: For $d = 5, 10, 20$ nodes projected to D = 100 dimensions, we evaluate our algorithm on synthetic ER-1, ER-2, and ER-4 DAGs, given the permutation. Figure 5 summarizes the results for 5 and 10 nodes. As in the linear case, the proposed method recovers the true causal structure and the true causal variables, and is significantly better than the VAE and GraphVAE baselines on all the metrics considered. For experiments in this setting, we noticed empirically that learning the permutation is hard, and performs not so different from a null graph baseline (Figure 13). This observation complements the identifiability result that recovery of latent variables is possible only upto a permutation in latent causal models (Brehmer et al., 2022; Liu et al., 2022) for general nonlinear mappings between causal variables and low-level data. This supports our observation of not being able to learn the permutation in nonlinear projection settings – but once the permutation is given to the model, it can quickly recover the SCM (figures 5,13). Refer figure 14 (in Appendix) for results on $d = 20$ nodes.

## 5.2 RESULTS ON LEARNING LATENT SCMs FROM PIXEL DATA

**Dataset and Setup**: A major challenge with evaluating latent causal discovery models on images is that it is hard to obtain images with corresponding GT graph and parameters. Other works (Kocaoglu et al., 2018; Yang et al., 2021; Shen et al., 2021) handle this by assuming the dataset is generated from certain causal variables (assumed to be attributes like gender, baldness, etc.) and a causal structure that is heuristically set by experts, usually in the CelebA dataset (Liu et al., 2015). This makes evaluation particularly noisy. Given these limitations, we verify if our model can perform latent causal discovery by evaluating on images from the chemistry dataset proposed in Ke et al. (2021) – a scenario where all GT factors are known. We use the environment to generate blocks of different intensities according to a linear Gaussian latent SCM where the parent block colors affect the child block colors then obtain the corresponding images of blocks. The dataset allows generating pixel data from random DAGs and linear SCMs. For this step, we use the same technique to generate causal variables as in the synthetic dataset section. Similar to experiments on nonlinear vector data, we are given the node ordering in this setting.

---

[2]An intervention set is defined as a set of nodes on which an intervention is performed

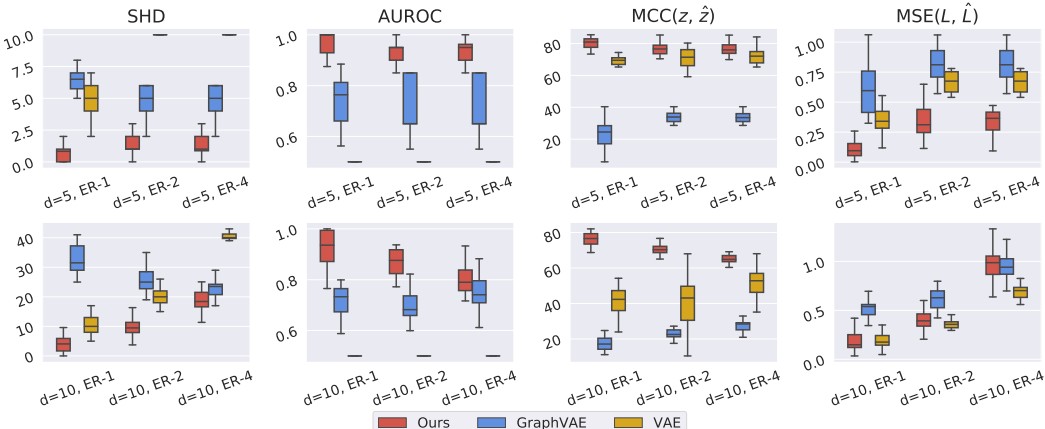

Figure 6: Learning the latent SCM from pixels of the chemistry dataset for $d = 5$ (top) and $d = 10$ nodes (bottom), given the node ordering. $\mathbb{E}$-**SHD** ($\downarrow$), **AUROC** ($\uparrow$), **MCC** ($\uparrow$), **MSE** ($\downarrow$)

**Results**: We perform experiments to evaluate latent causal discovery from pixels and known interventions. The results are summarized in figure 6. It can be seen that the proposed approach can recover the SCM significantly better than the baseline approaches in all the metrics even in the realistic dataset. In figure 7, we also assess the ability of the model to sample images from unseen interventions in the chemistry dataset by examining the generated images with GT interventional samples. The matching intensity of each block corresponds to matching causal variables, which demonstrates model generalization.

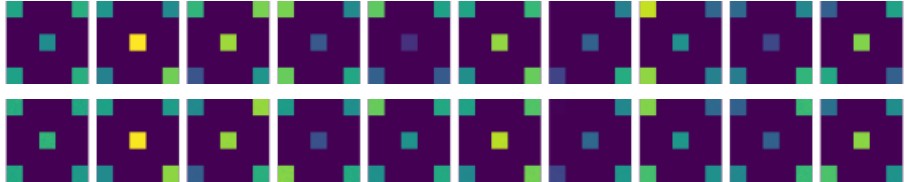

Figure 7: Image sampling from 10 random, unseen interventions: Mean over GT (top row) and predicted (bottom row) image samples in the chemistry dataset for $d = 5$ nodes.

## 6  LIMITATIONS

Our approach makes a number of assumptions. First, we assume the latent SCM is linear Gaussian. However, removing this restriction might be crucial to make the approach more general. Second, we assume access to the number of latent causal variables $d$. Extensions must consider how to infer the number of latent variables. We also make the assumption of known intervention targets whereas this might be restrictive for real-world applications. Future work could overcome this limitation by inferring interventions as in Hägele et al. (2022). Finally, we have assumed feasibility of interventions and known causal orderings for some of our experiments. However, in reality, some interventions could be infeasible and the causal ordering of latent variables might not be known.

## 7  CONCLUSION

We presented a tractable approximate inference technique to perform Bayesian latent causal discovery that jointly infers the causal variables, structure and parameters of linear Gaussian latent SCMs under random, known interventions from low-level data. The learned causal model is also shown to generalize to unseen interventions. Our Bayesian formulation allows uncertainty quantification and mutual information estimation which is well-suited for extensions to active causal discovery. Extensions of the proposed method to learn nonlinear, non-Gaussian latent SCMs from unknown interventions would also open doors to general algorithms that can learn causal representations.

## 8 REPRODUCIBILITY STATEMENT

The details regarding synthetic data generation of the DAGs, causal variables, and the high dimensional data is mentioned in section 5. The code for data generation, the models used, as well as for all the experiments is available at anonymous.4open.science/r/anon-biols-86E7. The average runtime of all experiments is documented in table 3. Appendix A.8 further evaluates experiments along additional metrics.

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

# A APPENDIX

## A.1 DERIVATION OF THE ELBO

We want to minimize the KL divergence between the true and approximate posterior:

$$D_{\mathrm{KL}}(q_\phi(\mathcal{Z}, \mathcal{G}, \Theta) \,||\, p(\mathcal{Z}, \mathcal{G}, \Theta \mid \mathcal{D}))$$
$$= -\mathbb{E}_{(\mathcal{G},\Theta)\sim q_\phi(\mathcal{G},\Theta)}\left[\mathbb{E}_{\mathcal{Z}\sim p(\mathcal{Z}|\mathcal{G},\Theta)}\left[\log p_\psi(\mathcal{D} \mid \mathcal{Z}, \mathcal{G}, \Theta)\right] - \log \frac{q_\phi(\mathcal{Z}, \mathcal{G}, \Theta)}{p(\mathcal{Z}, \mathcal{G}, \Theta)}\right] + \log p(\mathcal{D})\right]$$

where the prior and posterior factorize as (according to the explanation in 4.2):

$$p(\mathcal{Z}, \mathcal{G}, \Theta) = p(\mathcal{Z} \mid \mathcal{G}, \Theta) \cdot p(\mathcal{G}, \Theta) \tag{9}$$

$$q_\phi(\mathcal{Z}, \mathcal{G}, \Theta) = p(\mathcal{Z} \mid \mathcal{G}, \Theta) \cdot q_\phi(\mathcal{G}, \Theta) \tag{10}$$

Thus, we have that $D_{\mathrm{KL}}(q_\phi(\mathcal{Z}, \mathcal{G}, \Theta) \,||\, p(\mathcal{Z}, \mathcal{G}, \Theta \mid \mathcal{D}))$ reduces to:

$$-\mathbb{E}_{(\mathcal{G},\Theta)\sim q_\phi(\mathcal{G},\Theta)}\left[\mathbb{E}_{\mathcal{Z}\sim p(\mathcal{Z}|\mathcal{G},\Theta)}\left[\log p_\psi(\mathcal{D} \mid \mathcal{Z})\right] - \log \frac{q_\phi(\mathcal{G}, \Theta)}{p(\mathcal{G}, \Theta)}\right] + \log p(\mathcal{D})$$

$$= -\mathbb{E}_{(P,L,\Sigma)\sim q_\phi(P,L,\Sigma)}\left[\mathbb{E}_{\mathcal{Z}\sim p(\mathcal{Z}|P,L,\Sigma)}\left[\log p_\psi(\mathcal{D} \mid \mathcal{Z})\right] - \log \frac{q_\phi(P, L, \Sigma)}{p(P, L, \Sigma)}\right] + \log p(\mathcal{D})$$
$$\text{(from 8)}$$

$$= -\mathbb{E}_{(L,\Sigma)\sim q_\phi(L,\Sigma)}\left[\mathbb{E}_{P\sim q_\phi(P|L,\Sigma)}\left[\mathbb{E}_{\mathcal{Z}\sim p(\mathcal{Z}|P,L,\Sigma)}\left[\log p_\psi(\mathcal{D} \mid \mathcal{Z})\right] - \log \frac{q_\phi(P \mid L, \Sigma)}{p(P)}\right]\right.$$

$$\left. - \log \frac{q_\phi(L, \Sigma)}{p(L)p(\Sigma)}\right] + \log p(\mathcal{D}) \qquad \text{(via the factorization in 8)}$$

Since the log evidence $\log p(\mathcal{D})$ is a constant, minimizing the KL divergence corresponds to maximizing the following ELBO:

$$\max_{\phi,\psi} \mathbb{E}_{(L,\Sigma)\sim q_\phi(L,\Sigma)}\left[\mathbb{E}_{P\sim q_\phi(P|L,\Sigma)}\left[\mathbb{E}_{\mathcal{Z}\sim p(\mathcal{Z}|P,L,\Sigma)}\left[\log p_\psi(\mathcal{D}\mid\mathcal{Z})\right]-\log\frac{q_\phi(P\mid L,\Sigma)}{p(P)}\right.\right.$$
$$\left.\left.-\log\frac{q_\phi(L,\Sigma)}{p(L)p(\Sigma)}\right]\right]$$

## A.2 IMPLEMENTATION DETAILS

For all our experiments, we use the AdaBelief (Zhuang et al., 2020) optimizer with $\epsilon = 10^{-8}$ and a learning rate of 0.0008. $\tau$ is set to 0.2. Our experiments are fairly robust with respect to hyperparameters and we did not perform hyperparameter tuning for any of our experiments. Table 1 and 2 summarizes the network details for $\text{MLP}_{\phi(T)}$ and the decoder $p_\psi(\mathcal{X}\mid\mathcal{Z})$.

Table 1: Network architecture for $\text{MLP}_{\phi(T)}$

| Layer type | Layer output | Activation |
|---|---|---|
| Linear | 128 | GeLU |
| Linear | 128 | GeLU |
| Linear | $d^2$ | |

Table 2: Network architecture for the decoder $p_\psi(\mathcal{X}\mid\mathcal{Z})$

| Layer type | Layer output | Activation |
|---|---|---|
| Linear | 16 | GeLU |
| Linear | 64 | GeLU |
| Linear | 64 | GeLU |
| Linear | 64 | GeLU |
| Linear | D | |

## A.3 TRAINING CURVES

Figures 8, 9, and 10 summarize the training curves for $d = 6, 20, 50$ nodes on ER-DAGs of degree 1, 2, and 4. Each plot contains 3 lines that shows training with observational, single-interventional, and multi-interventional data.

## A.4 LINEAR PROJECTION EXPERIMENTS

Details for figure 11, $d = 10$ nodes: The dataset consists of 500 observational points and 20000 interventional points. To sample the 20000 interventional points, we randomly choose 200 intervention sets, and for each intervention set we sample 100 data points. The model was trained for 8000 epochs to reach convergence.

Details for figure 12, $d = 20$ nodes: The dataset consists of 500 observational points and 20000 interventional points. To sample the 20000 interventional points, we randomly choose 100 intervention sets, and for each intervention set we sample 200 data points. The model was trained for 3000 epochs to reach convergence.

## A.5 NONLINEAR PROJECTION EXPERIMENTS

Figure 13 contains the results for the latent causal discovery problem with and without learning a permutation. Figure 14 shows the results for 500 observational samples and 10000 interventional

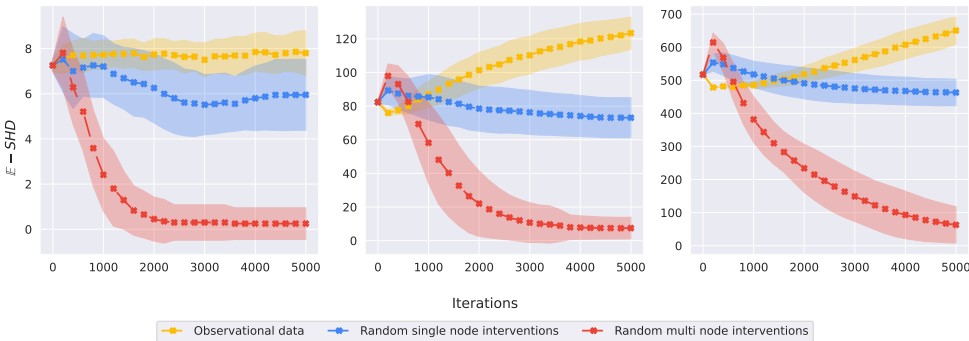

Figure 8: Learning latent SCM parameters given a fixed node ordering for linearly projected causal variables for random ER-1 DAGs with $d = 6, 20, 50$ nodes. The model was trained for 5000 iterations over 3500 data samples out of which 500 were observational points for the single and multi node intervention runs.

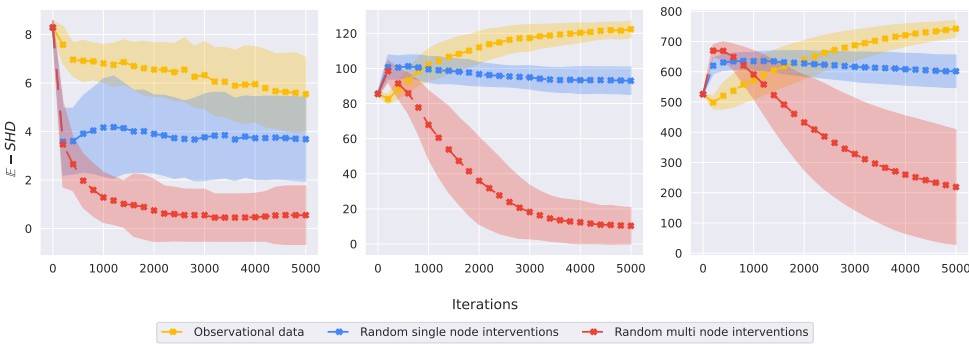

Figure 9: Learning latent SCM parameters given a fixed node ordering for linearly projected causal variables for random ER-2 DAGs with $d = 6, 20, 50$ nodes. The model was trained for 5000 iterations over 3500 data samples out of which 500 were observational points for the single and multi node intervention runs.

samples with 100 intervention sets and 100 samples per set on $d = 20$ nodes and $D = 100$ dimensions.

## A.6 ABLATION STUDY: EFFECT OF PERFORMANCE WITH RESPECT TO NUMBER OF INTERVENTION TYPES

Here, we study how the performance and recovery of the latent SCM is affected with respect to the number of intervention types in the dataset. The number of intervention types refers to different combinations of nodes we perform interventions on. For all experiments in this subsection, we use 100 interventional samples per type of intervention. Figure 15 and 16 show results on vector data where the high-dimensional vector is a linear projection of the causal variables. Figure 17 and 18 summarize results on vector data where the high-dimensional vector is a nonlinear projection of the causal variables.

## A.7 ADDITIONAL RELATED WORK

**Latent variable models with predefined structure**: Examples include VAE (Kingma and Welling, 2013; Rezende et al., 2014) which has an independence assumption between latent variables. To overcome this, Sønderby et al. (2016) and Zhao et al. (2017) define latent variables with a chain structure in VAEs. Kingma et al. (2016) uses inverse autoregressive flows to improve upon the diagonal covariance of latent variables in VAEs.

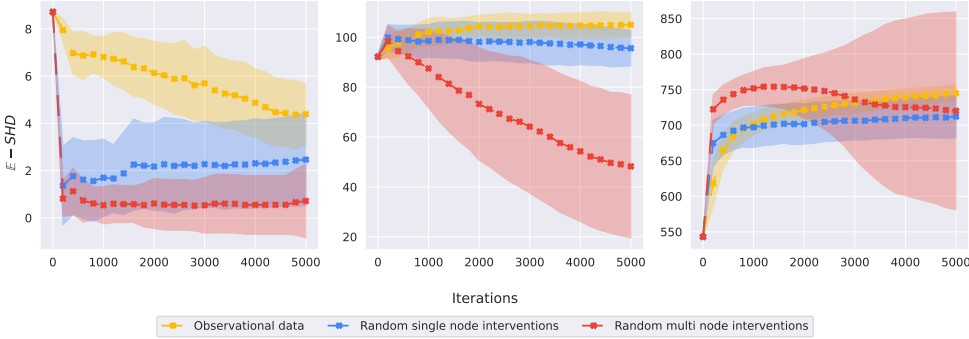

Figure 10: Learning latent SCM parameters given a fixed node ordering for linearly projected causal variables for random ER-4 DAGs with $d = 6, 20, 50$ nodes. The model was trained for 5000 iterations over 3500 data samples out of which 500 were observational points for the single and multi node intervention runs.

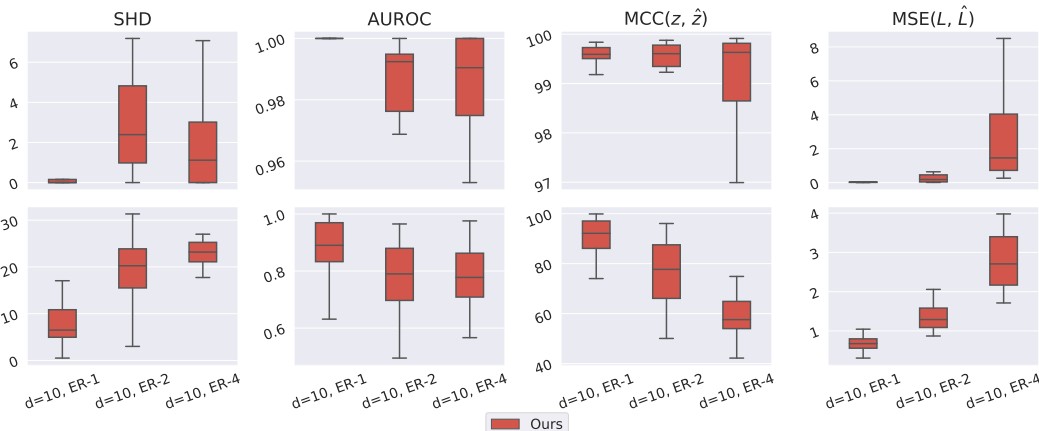

Figure 11: Learning the latent SCM (i) given a node ordering (top) and (ii) over node orderings (bottom) for **linear** projection of causal variables for $d = 10$ nodes, D = 100 dimensions.

**Causal discovery**: Annadani et al. (2021) casts the Bayesian structure learning problem as an autoregressive one by sequentially predicting edges, in hopes of capturing the potentially multi-modal posterior. Deleu et al. (2022) uses Generative Flow Networks, or GFlowNets (Bengio et al., 2021), a new class of probabilistic methods that lies at the intersection of reinforcement learning and variational inference. The work uses the transitive closure property ensuring that the action space is constrained to actions that do not introduce cycles. Chickering (2002) proposes a greedy search algorithm, but does not scale to large number of nodes. Wang et al. (2022) leverages sum product networks to perform exact Bayesian structure learning. Hägele et al. (2022) extends the framework of Lorch et al. (2021) to perform Bayesian causal discovery in a setting where interventions are unknown. Xie et al. (2022) is in a setting where the edges exist not just between latent causal variables but with high-dimensional variables in the dataset as well. Other efforts include (Shimizu et al., 2011; Lopez-Paz and Oquab, 2016; Yu et al., 2019; Ghoshal and Honorio, 2018; Ng et al., 2020; Li et al., 2022).

## A.8    COMPLETE EVALUATION

For the experiments already presented in the main text, this section contains additional a more comprehensive evaluation on the following metrics:

- **SHD_C**: The expected CPDAG SHD between the GT and predicted DAG's skeletons.

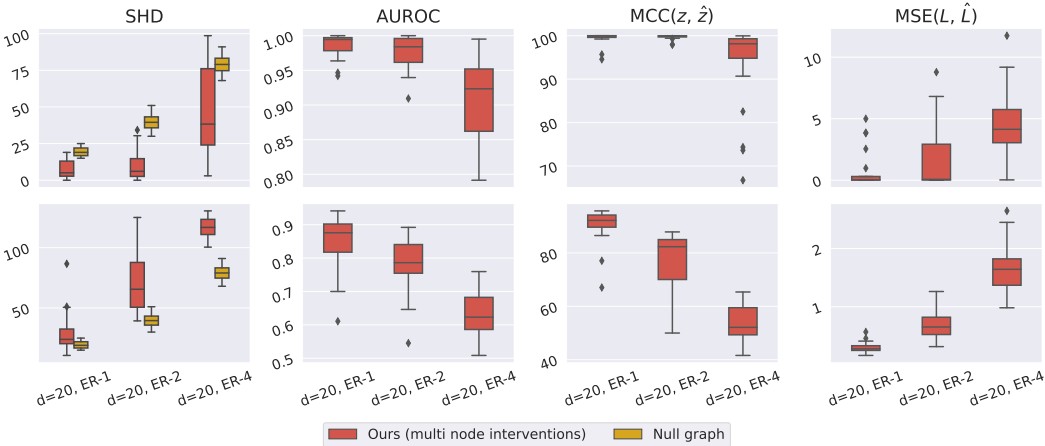

Figure 12: Learning the latent SCM (i) given a node ordering (top) and (ii) over node orderings (bottom) for **linear** projection of causal variables for $d = 20$ nodes, D $= 100$ dimensions.

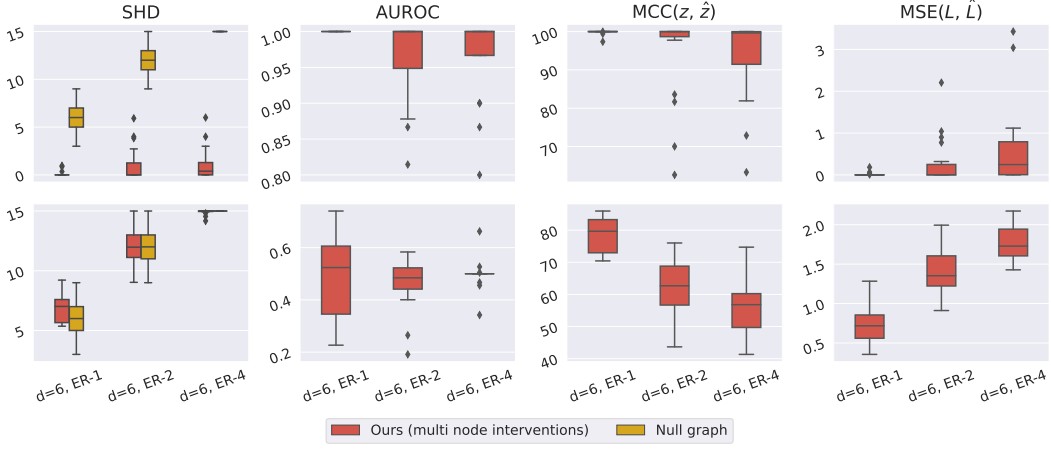

Figure 13: Learning the latent SCM (i) given a node ordering (top) and (ii) over node orderings (bottom) for **nonlinear** projection of causal variables for $d = 6$ nodes, D $= 100$ dimensions.

- **SHD**: The expected SHD between the GT and predicted DAG.
- **MCC**: The expected Mean Correlation Coefficient between the predicted and true causal variables.
- **AUROC** between predicted and true graph structure.
- **FPR**: False positive rate
- **FN**: False negative
- **TP**: True positive
- **AUPRC_G**: Area under precision-recall curves
- **Precision** of the graph structure prediction
- **TN**: True negatives
- **FP**: False positives
- **TPR**: True positive rate
- **Recall** of the graph structure prediction
- **AUPRC_W**:
- **F1 Score** $= 2 * Precision * Recall/(Precision + Recall)$
- **L_MSE**: Mean squared error between predicted and true edge weights.

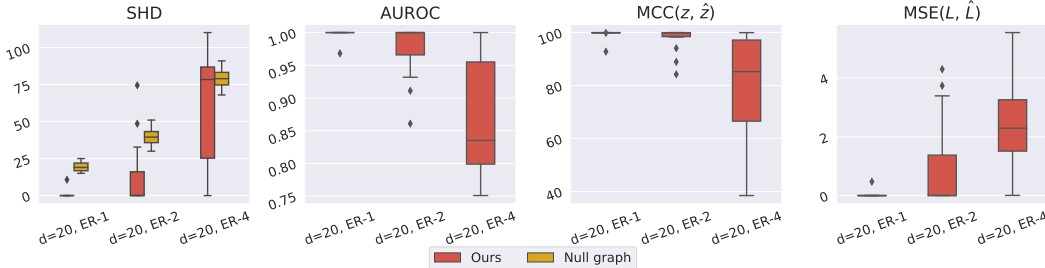

Figure 14: Learning the latent SCM given a node ordering for **nonlinear** projection of causal variables for $d = 20$ nodes, D $= 100$ dimensions.

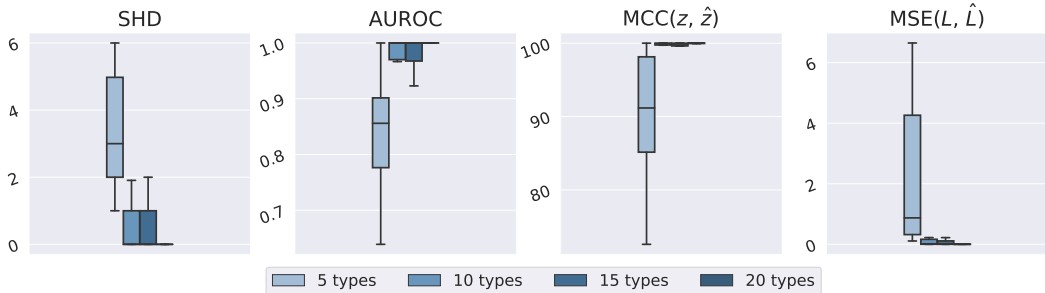

Figure 15: Effect of number of intervention types seen versus on the performance of learning the latent SCM given a node ordering for $d = 5$ nodes, D $= 100$ dimensions.

- **X_MSE**: Mean squared reconstruction error over the high dimensional (low-level) data
- **true_obs_KL_term_Z**: The KL divergence between the predicted and GT observational joint distributions.

Figure 19 shows the complete evaluation on 18 different metrics for $d = 5$ nodes. The dataset consists of 500 observational points and 2000 interventional points. To sample the 2000 interventional points, we randomly choose 20 intervention sets, and for each intervention set we sample 100 data points with random intervention values.

## A.9 DISCUSSION ON IDENTIFIABILITY

Identifiability is an important topic of discussion when making conclusions about the recovering the structure of a causal model. However in our work, we approximate a full posterior distribution over the latent SCMs, instead of returning just a single graph. In this setting, questions of identifiability become less critical, as we can assign probabilities for many possible candidate graphs (and parameters) to express our level of confidence that a particular SCM yields the correct causal conclusions. This is a softer guarantee than what identifiability would provide.

Nevertheless, we refer the reader to recent works in causal representation learning (Brehmer et al., 2022; Ahuja et al., 2022a;c), that have given some identifiability guarantees in conditions similar to ours. Particularly, we rely on the identifiability results presented in Brehmer et al. (2022). *But identifiability does not imply learnability*. Thus, in this work, we are concerned with the problem of how one can devise a principled practical algorithm to learn a distribution over latent SCMs.

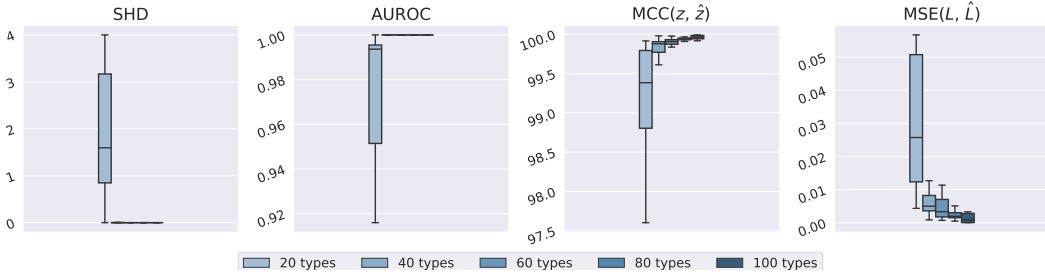

Figure 16: Effect of number of intervention types seen versus on the performance of learning the latent SCM given a node ordering for $d = 10$ nodes, D = 100 dimensions.

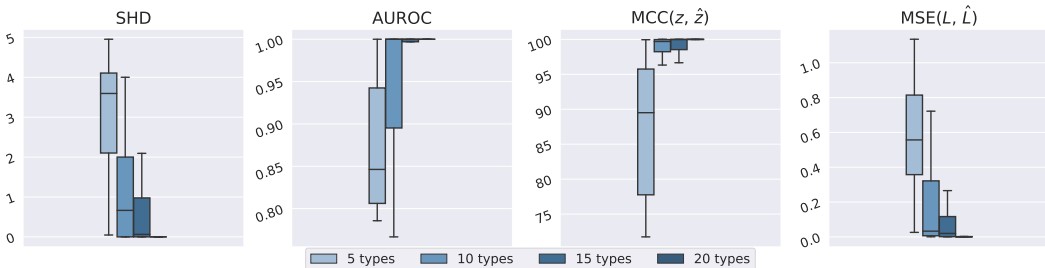

Figure 17: Ablation: effect of number of intervention types seen versus on the performance of learning the latent SCM given a node ordering for $d = 5$ nodes, D = 100 dimensions.

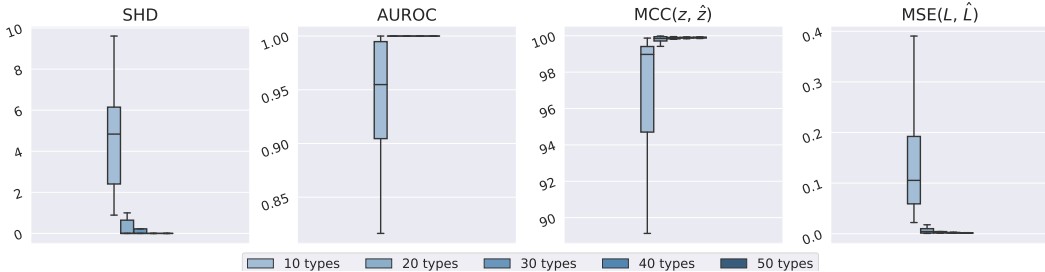

Figure 18: Ablation: effect of number of intervention types seen versus on the performance of learning the latent SCM given a node ordering for $d = 10$ nodes, D = 100 dimensions.

Table 3: Average experiment runtimes

| Nodes ($d$) | Dataset size | Experiment | Projection | Steps | Avg. runtime (min) |
|---|---|---|---|---|---|
| 5 | 2500 | Vector data | Linear (fixed ordering) | 5000 | 2 |
| 10 | 10500 | Vector data | Linear (fixed ordering) | 3000 | 6 |
| 10 | 10500 | Vector data | Linear (learned ordering) | 3000 | 60 |
| 20 | 80500 | Vector data | Linear (fixed ordering) | 3000 | 150 |
| 20 | 20500 | Vector data | Linear (learned ordering) | 3000 | 360 |
| 20 | 80500 | Vector data | Linear (learned ordering) | 8000 | 540 |
| 10 | 5500 | Vector data | Nonlinear | 2000 | 12 |
| 5 | 2500 | Vector data | Nonlinear | 5000 | 15 |
| 20 | 10500 | Vector data | Nonlinear | 10000 | 100 |
| 5 | 2500 | Image data | Nonlinear | 2000 | 40 |
| 5 | 5500 | Image data | Nonlinear | 2000 | 75 |
| 10 | 2500 | Image data | Nonlinear | 2000 | 50 |
| 10 | 5500 | Image data | Nonlinear | 2000 | 80 |

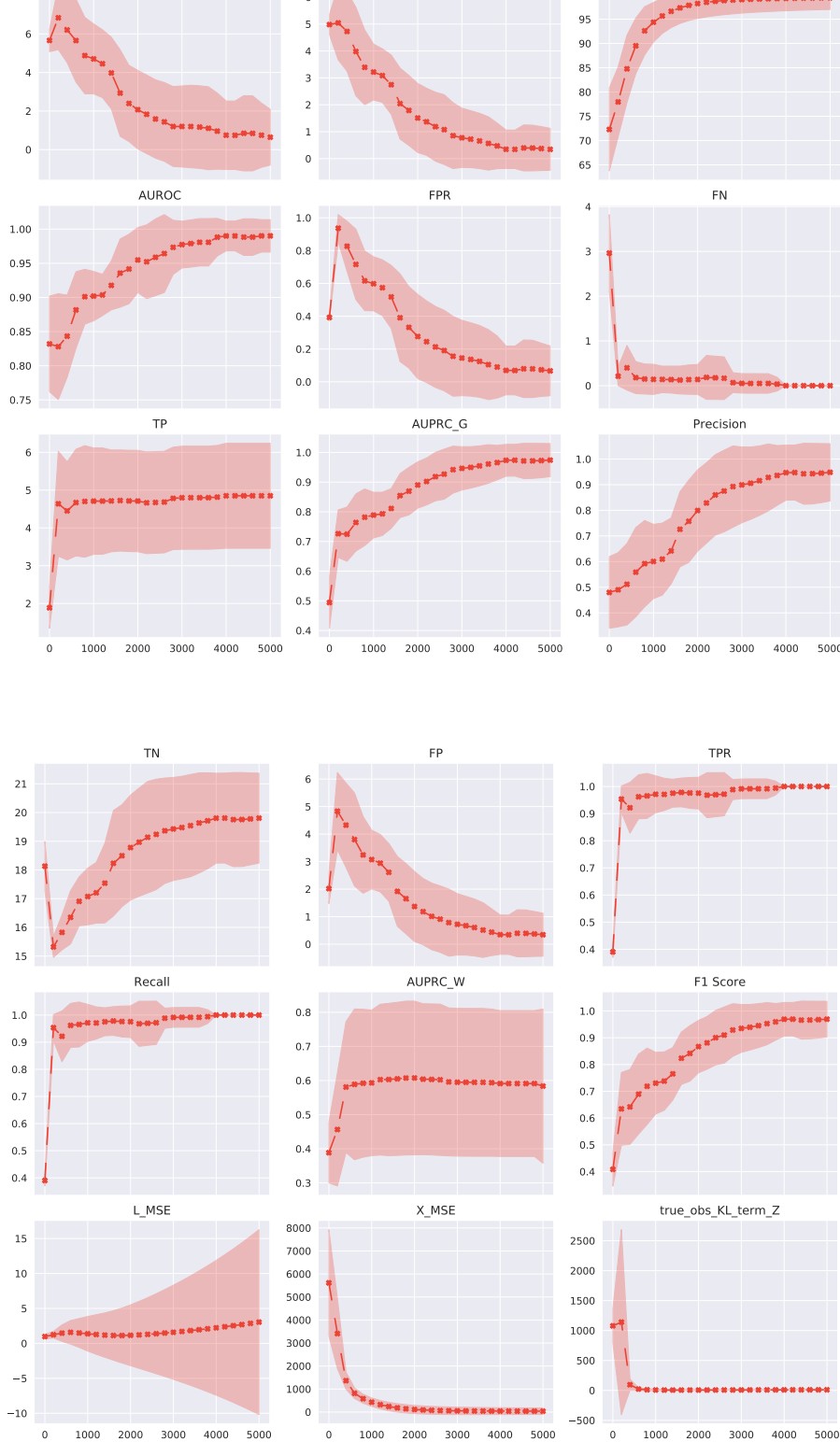

Figure 19: Training curves: Metric versus number of iterations for $d = 5$ nodes linearly projected to $D = 100$ dimensions, with 20 intervention sets, 100 interventional samples per intervention set, and 500 observational samples.

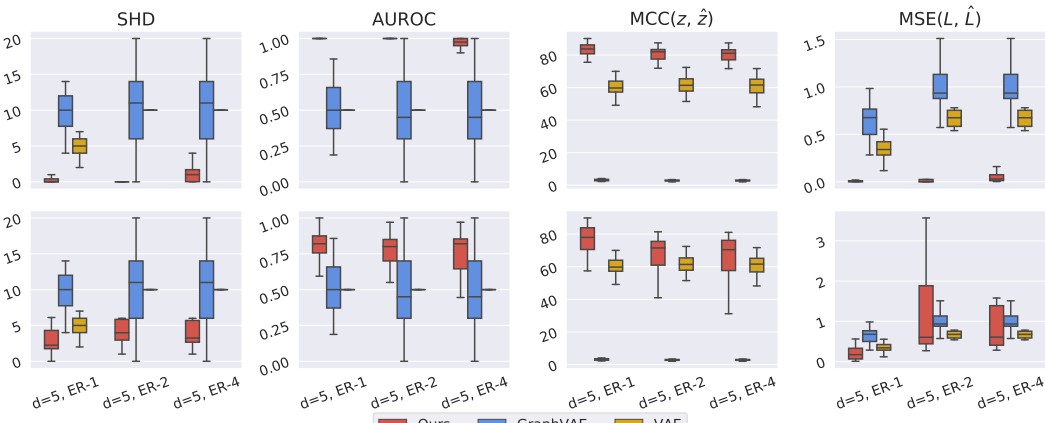

Figure 20: Learning the latent SCM (i) given a node ordering (top) and (ii) over node orderings (bottom) for **linear** projection of causal variables for $d = 5$ nodes, $D = 100$ dimensions with non-equal noise variance, stochastic decoder, and 10000 iterations: $\mathbb{E}$-**SHD** ($\downarrow$), **AUROC** ($\uparrow$), **MCC** ($\uparrow$), **MSE** ($\downarrow$) [updated with non-equal noise variance, stochastic observation generating process, and stochastic decoder].

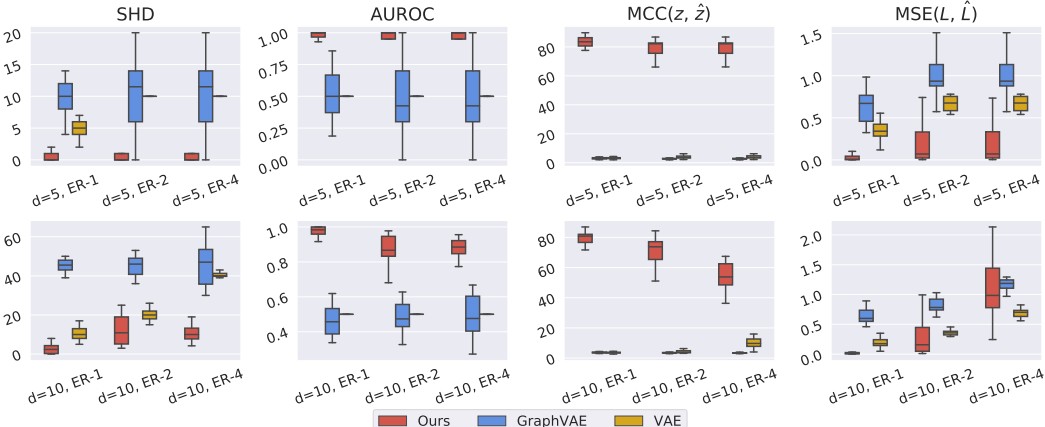

Figure 21: Learning the latent SCM for **nonlinear** projection of causal variables for $d = 5$ (top) and $d = 10$ (bottom) nodes, $D = 100$ dimensions, given the node ordering. $\mathbb{E}$-**SHD** ($\downarrow$), **AUROC** ($\uparrow$), **MCC** ($\uparrow$), **MSE** ($\downarrow$) [updated with non-equal noise variance, stochastic observation generating process, and stochastic decoder].

