# OpenReview forum: "Learning Latent Structural Causal Models"
_ICLR.cc/2023/Conference — Submitted to ICLR 2023_

### Official Review · Reviewer_VekW · 2022-10-18

**Confidence:** 4
**Correctness:** 4
**Technical Novelty And Significance:** 3
**Empirical Novelty And Significance:** 3
**Recommendation:** 8

**Clarity, Quality, Novelty And Reproducibility:**

The content of the paper clear and is self contained (as much as possible). But for the comments already provided, the paper is of good quality. The main concern is on motivating the value of discovering (if any exists) a latent, hardly interpretable, SCM in practical applications.
The work is novel, in particular in respect to the learning of a *latent* causal model.
Code is provided but it is unclear how one should use it to reproduce results (however I believe the paper refers to models 6 and 7, vae-dibs and decoder-dibs respectively)

**Strength And Weaknesses:**

The paper provide a flexible formulation for latent structural causal models, with potential extensions beyond the case of linear Gaussian ones.

Here are some recommendations that might improve the manuscript.
1) The scope of the work should be better framed, for a wider audience. In the examples presented the ground truth was present. In practical applications there may not be a ground truth. It should be made clear what are the benefits of discovering a latent SCM in this practical context, where the SCM may also lack interpretability. What are the benefits in these cases?
2) It is not clear how to pick d, the dimension of the latent space. In the examples provided, it seems d is always picked a priori to be the correct one. In practical applications this could not be possibile. It is then necessary to know what are the consequences of under/over estimating d.
3) But for the algorithm listing, it is not explained how the interventions enter the general method. I would recommend to elaborate on that.
4) When explaining the distribution of P, and in the algorithm, it is not clear whether the temperature parameter \tau gets updated through the epochs. It seems it is kept fixed and the Hungarian algorithm takes care of \tau -> 0. I would make it explicit.
5) Evaluation metrics: the metrics focus on the recovery of the latent SCM, while the main application would be improved sample generation of the observed variables. It would be beneficial to measure the generative power of the learned model. This is done only qualitatively in Figure 7. It would be good to have also a quantitative assessment.
6) The plot of the MSE while learning the node orderings in figure 4 shows a higher MSE for the proposed method versus the other. You should elaborate on that.
7) For both of the examples the dimension of the latent space is known?
8) It should be made explicit whether the results in figure 6 are with known P or not.


**Summary Of The Paper:**

The paper presents a novel technique for the discovery and characterization of latent Gaussian linear structural causal models.
It employs a Bayesian formulation of the problem and solve it in variational form.
Finally it provides performance examples using synthetic data in a purely ideal case and a case of imaging for chemistry. Due to the lack of competing methodologies, results are compared with marginally independent VAE and Graph VAE, neither of which tackle exactly the same task.

**Summary Of The Review:**

Overall a good paper, which needs a better framing to be appealing to a broader audience.

---

> ### Author Response · Authors · 2022-11-15
> **Response to reviewer VekW (1/2)**
>
> Thanks a lot for your comments and overall feedback!
>
> > 4. When explaining the distribution of P, and in the algorithm, it is not clear whether the temperature parameter \tau gets updated through the epochs. It seems it is kept fixed and the Hungarian algorithm takes care of \tau -> 0. I would make it explicit.
>
> We have now made this more explicit in the paper (section 4.3 paragraph on **Distribution over P**): "In the limit of infinite iterations and as $\tau \to 0$, sampling from the distribution returns a doubly stochastic matrix. During the forward pass, a hard permutation $P$ is obtained by using the Hungarian algorithm \citep{hungarian} which allows $\tau \to 0$.". We have also made this explicit in Algorithm 1, line 7.
>
> > 6. The plot of the MSE while learning the node orderings in figure 4 shows a higher MSE for the proposed method versus the other. You should elaborate on that.
>
> Thanks for pointing out this. The MSE between ground truth and predicted edge weights is just a proxy metric for how well our method is performing and is *not a perfect metric*. Moreover, this metric makes sense only for models that learn/infer the parameters (or edge weights for linear SCMs), such as ours, and neither of the baselines have the ability to infer parameters. We however included the MSE between ground truth and predicted edge weights for the sake of completeness.
>
> Nevertheless, we speculated that the reason for higher MSE is that learning the permutation (aka node ordering) in the experiment (bottom row of Fig. 4), makes it a harder problem compared to other experiments. **We suspected that either more data or longer training would fix this issue**. To confirm this suspicion, we allowed the algorithm to run for more iterations and we see that MSE($L$, $\hat{L}$) do in fact get better with longer training and now performs better than both baselines ([figure 20](https://ibb.co/YTgH25s) of the appendix).
>
> To address yours and other reviewers’ comments (jaMU, Aq59, oCT2), and also to remove some of our assumptions, we made the following changes to the experimental setup:
> - Moved from an equal-noise variance setting to non-equal noise variance setting which is more general.
> - Earlier, the observation generating function mapping from latent to observed variables was deterministic. We have now made this stochastic and also use a stochastic decoder now.
> - Train for longer with 10k iterations instead of the earlier 5k iterations.
>
> With all these changes done (all hyperparameters and other experimental conditions were kept constant), we re-ran the experiment for figure 4. We note that our method continues to perform best among baselines and that, as suspected, **longer training  does improve performance and causes the MSE($L$, $\hat{L}$) of the proposed method to outperform baselines**. We will replace figure 4 with this image soon in a future revision, but for now, we add the image to the end of the appendix (please see figure 20).
>
> [Figure 20](https://ibb.co/YTgH25s) in the appendix shows same experiment as in [figure 4](https://ibb.co/0cPT6j3) (linear projection of vector data) with non-equal noise variance.
>
> [Figure 21](https://ibb.co/VBpmS1M) in the appendix shows same experiment as in [figure 5](https://ibb.co/SxsGvVH) (nonlinear projection of vector data) with non-equal noise variance and stochastic observation generating function.
>
> > 7. For both of the examples the dimension of the latent space is known?
>
> Yes, in this work, we assume access to number of latent nodes and study the scenario where each latent node is a scalar. However, in future work we plan to consider the case where the number of nodes and each node dimension are unknown.
>
> > 8. It should be made explicit whether the results in figure 6 are with known P or not.
>
> Thanks for pointing this out, we have made this explicit in the figure title as well as in the text of section 5.2.
>
> Regarding point 5, do you have any suggestions on what metrics would be relevant?
>
> **Other changes**: (i) we have corrected some notations throughout the text, (ii) introduced an additional paragraph "Structure learning with latent variables" in the related works (section 3) where we have discussed more works, (iii) included a new limitations section (section 6) and (iv) a new subsection in the appendix (A.2) for implementation details and (v) a new subsection (A.9) that discusses identifiability in our setting.
>
> We will put up more comments to address the rest of your concerns (for 1, 2, 3, 5). Meanwhile, if you have any comments or further things to be fixed with regards to points 4, 6, 7 or 8, do let us know and we would be glad to fix this.

---

> ### Author Response · Authors · 2022-12-07
> **Discussion period coming to an end; any remaining concerns?**
>
> Dear Reviewer VekW,
>
> Thanks a lot for your insightful comments and feedback in the review! This is a gentle reminder that the discussion period is coming to an end. We have addressed your concerns in our responses and we would like to know if there are any remaining concerns?

---

> > ### Comment · Reviewer_VekW · 2022-12-07
> > **Response**
> >
> > Thanks, you addressed my main concers.
> >
> > Regarding point 5, I was wondering whether posterior predictive checks could be used to asses the quality of the posterior distribution learned.

---

### Official Review · Reviewer_Aq59 · 2022-10-20

**Confidence:** 3
**Correctness:** 2
**Technical Novelty And Significance:** 2
**Empirical Novelty And Significance:** 2
**Recommendation:** 3

**Clarity, Quality, Novelty And Reproducibility:**

 It makes some progress on the hard problem of learning the latent causal structures of interest from the ﻿unstructured dataset.

**Strength And Weaknesses:**

Strength

The authors focus on the challenge of causal representation learning. This is an important but challenging problem. While some methods have been proposed, they may incorporate some prior information or restrict the latent variable to be independent.

The authors designed a ﻿simple factorization of the posterior to learn the structure of latent variables.

Weakness

The authors do not analyze the identification of their model (linear latent SCM). Lack of identifiability conclusions may not guarantee the correctness of the results. This is an important step before learning the structural causal model. The following related works may be useful for it.


Ricardo Silva, Richard Scheine, Clark Glymour, and Peter Spirtes. Learning the structure of linear
latent variable models. Journal of Machine Learning Research, 7(Feb):191–246, 2006.

Erich Kummerfeld and Joseph Ramsey. Causal clustering for 1-factor measurement models. In
Proceedings of the 22nd ACM SIGKDD International Conference on Knowledge Discovery and
Data Mining, ACM, 2016.

Feng Xie, Ruichu Cai, Biwei Huang, Clark Glymour, Zhifeng Hao, and Kun Zhang. Generalized
independent noise condition for estimating latent variable causal graphs. NeurIPS,  2020.

Bohdan Kivva, Goutham Rajendran, Pradeep Ravikumar, and Bryon Aragam. Learning latent causal
graphs via mixture oracles. NeurIPS, 2021.


The model requires that all noise terms of latent variables have equal noise variance.

The authors need to state the number of latent variables is known in advance in their setup.



**Summary Of The Paper:**

This paper studies latent causal structure learning from ﻿unstructured, low-level, observations. The authors first formulate the learning problem into the Bayesian problem and use the variational inference to estimate the joint posterior over the latent variables. Then the authors apply their methods in both synthetic and pixel datasets to verify the efficiency.

**Summary Of The Review:**

The problem is challenging and the authors design a new method to address it. However, there is no theoretical results to support their method.

---

> ### Author Response · Authors · 2022-11-12
> **Response to reviewer Aq59 (2/2)**
>
> > 2. The model requires that all noise terms of latent variables have equal noise variance.
>
> To overcome this assumption, **we have extended our experiments to incorporate non-equal noise variances and present experimental results for this setting as well**.
>
> To address yours and other reviewers’ comments (jaMU), and also to remove some of our assumptions, we made the following changes to the experimental setup:
>
> - Moved from an equal-noise variance setting to non-equal noise variance setting which is more general.
> - Earlier, the observation generating function mapping from latent to observed variables was deterministic. We have now made this stochastic and also use a stochastic decoder now.
> - Train for longer with 10k iterations instead of the earlier 5k iterations.
>
>
> Under these experimental conditions we re-ran our method as well as the VAE and GraphVAE baselines. All hyperparameters and other experimental conditions were kept constant. We note that **our method continues to outperform baselines even if these assumptions** (equal noise variance, deterministic decoder and observation generation function) **are removed**. We also notice that the MSE($L$, $\hat{L}$) between GT and inferred parameters (2nd row, last column of figure 4) get better with respect to baselines, due to longer training.
>
> [Figure 20](https://ibb.co/YTgH25s) in the appendix shows same experiment as in [figure 4](https://ibb.co/0cPT6j3) (linear projection of vector data) with non-equal noise variance.
>
> [Figure 21](https://ibb.co/VBpmS1M) in the appendix shows same experiment as in [figure 5](https://ibb.co/SxsGvVH) (nonlinear projection of vector data) with non-equal noise variance.
>
> If you think there are any other unaddressed concerns, please let us know and we would be happy to run them.
>
> > 3. The authors need to state the number of latent variables is known in advance in their setup.
>
> It is true that we assume the number of latent variables to be known. But we would like to point out that **this is a common assumption made in the literature** -- all the baselines (VAE and GraphVAE) and all prior works in causal representation learning have the same assumption (e.g., [1-3]). More research is required to address the case where the number of latent variables is not known apriori, as this would mean that the Bayesian causal discovery module needs to be able to deal with predicting the number of causal variables, and is a significantly harder problem. We believe that treatment of such a problem **merits a separate work on its own** and is beyond the current scope of this paper as well as most current causal representation learning methods. However, we have noted this assumption of ours under a new “limitations” section in our latest revision (section 6).
>
> [1] Brehmer, J., Haan, P. De, Lippe, P. & Cohen, T. Weakly supervised causal representation learning. in *ICLR2022 Workshop on the Elements of Reasoning: Objects, Structure and Causality* (2022).
>
> [2] Mengyue Yang, Furui Liu, Zhitang Chen, Xinwei Shen, Jianye Hao, and Jun Wang. Causalvae: Disentangled representation learning via neural structural causal models. In Proceedings of the IEEE/CVF Conference on Computer Vision and Pattern Recognition, pages 9593–9602, 2021.
>
> [3] Xinwei Shen, Furui Liu, Hanze Dong, Qing Lian, Zhitang Chen, and Tong Zhang. Disentangled generative causal representation learning, 2020

---

> ### Author Response · Authors · 2022-11-12
> **Response to reviewer Aq59 (1/2)**
>
> Thanks for your overall feedback and comments on our work!
>
> > 1. The authors do not analyze the identification of their model (linear latent SCM). Lack of identifiability conclusions may not guarantee the correctness of the results. This is an important step before learning the structural causal model. These works may be useful for it.
>
> - We agree that questions of identifiability are important when making conclusions about the structure of a causal model, for methods returning only a single structure (”maximum likelihood methods”, as mentioned in our Related Work). However in our work, we have access to interventional data and approximate a full posterior distribution over the latent SCMs, instead of returning just a single graph. In this setting, questions of identifiability become less critical, as we can assign probabilities for many possible candidate graphs (and parameters) to express our level of confidence that a particular SCM yields the correct causal conclusions. This is a softer guarantee than what identifiability would provide. That being said, we would like to note that **recent prior work on causal representation learning [1,2,3] has addressed these questions of identifiability under similar conditions as ours** (identifying latent causal models from low-level observations), but again more applicable to “maximum-likelihood methods” returning a single graph.
> - Also, thanks for pointing out the papers. While these works are relevant and have been cited in our latest revision, we would like to point out that these papers are in a different setting where some variables of the causal model are observed and other variables are unobserved. Moreover, these works consider causal edges between the latent and observed variables. On the other hand, our setup involves a set of latent variables which explain some low-level high dimensional data, and a causal model is defined over the entire set of latent variables. There are no causal edges between latent variables and the low-level data.
> - Our setup is similar to [1] and in fact, our proposed method relies on the identifiability result of [1]. We have also acknowledged this fact in the results: **Section 5.1.2 paragraph on Results on nonlinear projection of causal variables**, “*This observation complements the identifiability result that recovery of latent variables is possible only upto a permutation in latent causal models [1,4] for general nonlinear mappings between causal variables and low-level data*”.
> - For completeness and to further aid the clarity of the paper, we have also **included a** **discussion on identifiability** in our revision under a new, separate section A.9 (”Discussion on identifiability”) of the appendix as well cited these works in the last paragraph of the related works section.
>
> [1] Brehmer, J., Haan, P. De, Lippe, P. & Cohen, T. Weakly supervised causal representation learning. in *ICLR2022 Workshop on the Elements of Reasoning: Objects, Structure and Causality* (2022).
>
> [2] Kartik Ahuja, Jason Hartford, & Yoshua Bengio. Weakly supervised representation learning with sparse perturbations.
>
> [3] Kartik Ahuja, Divyat Mahajan, Vasilis Syrgkanis, & Ioannis Mitliagkas. Towards efficient representation identification in supervised learning.
>
> [4] Yuhang Liu, Zhen Zhang, Dong Gong, Mingming Gong, Biwei Huang, Anton van den Hengel, Kun Zhang, and Javen Qinfeng Shi. Weight-variant latent causal models, 2022.
>
> Please let us know if you have further concerns in this regard and we would be happy to address them!

---

> ### Author Response · Authors · 2022-11-24
> **Further questions to be addressed?**
>
> Dear Reviewer Aq59,
>
> Thank you again for your review. We believe we have addressed all concerns. Please let us know if you have further questions so we can continue the discussion and see if there are any concerns left. If there are no further questions, we would be happy if you could consider raising the score.

---

> ### Author Response · Authors · 2022-12-07
> **Discussion period coming to an end; any remaining concerns?**
>
> Dear Reviewer Aq59,
>
> This is a gentle reminder that the discussion period is coming to an end. We have addressed all concerns brought up in our detailed responses and we would like to know if there are any remaining concerns? If there are no remaining concerns, adjusting the rating accordingly would be helpful.

---

> > ### Comment · Reviewer_Aq59 · 2022-12-07
> > **One question**
> >
> > Thanks for the author's detailed response.  The authors say the identifiability of the model has been discussed by other published papers. After reading Ref [1],  it seems not to be easy to show that the model of interest is still identifiable. In Ref [1], the causal mechanisms  f_i(.;z_pa_i) is invertible, differentiable, and its inverse is differentiable. However,  in the current linear additive noise SCMs, it seems not. Please correct me if I understand something wrong.
> >
> > [1] Brehmer, J., Haan, P. De, Lippe, P. & Cohen, T. Weakly supervised causal representation learning. in ICLR2022 Workshop on the Elements of Reasoning: Objects, Structure and Causality (2022).

---

> > > ### Author Response · Authors · 2022-12-09
> > > **Causal mechanisms are invertible, differentiable, and the inverse is differentiable as well**
> > >
> > > Thanks for raising the question. Please refer page 3 of [Ref [1]](https://arxiv.org/abs/2203.16437) which states:
> > >
> > > > An SCM entails a unique solution s : $\varepsilon$ → $\mathcal{Z}$ defined by successively applying the causal mechanisms. We require the causal mechanisms to be pointwise diffeomorphic, that is, for any value of the parents $z_{pa_i}$ we have that $f_i(·; z_{pa_i})$ is invertible, differentiable, and its inverse is differentiable.
> > >
> > > The footnote at the end of the same page includes the following statement which clarifies the question:
> > > > Under some mild smoothness assumptions, any SCM can be brought into this form by elementwise redefinitions of the variables, preserving the observational and interventional distributions.
> > >
> > > However, we also show this concretely for our linear Gaussian additive noise SCM case. Consider $f_i(\varepsilon_i; z_{pa_i})$ which locally maps $\varepsilon_i \rightarrow \mathcal{Z}_i$ according to:
> > >
> > > $\sum\limits_{p \in {pa_i}} w_{ip} z_p + \varepsilon_i$
> > >
> > > - $f_i(\varepsilon_i; z_{pa_i})$ is a linear function and is differentiable with respect to $\varepsilon_i$
> > > - The inverse $g_i(.; z_{pa_i}) = f_i^{-1}(.; z_{pa_i})$ exists and maps $\mathcal{Z}_i \rightarrow \varepsilon_i$ according to:
> > >
> > > $\sum\limits_{p \in i \cup pa_i } \widetilde{w}_{ip} z_p$ where
> > >
> > > $\widetilde{w}_{ip} = 1$ if $p=i$ and
> > >
> > > equal to $-w_{ip}$ otherwise
> > > - The inverse $g_i(.; z_{pa_i}) = f_i^{-1}(.; z_{pa_i})$ is also a linear function and is differentiable with respect to $z_i$
> > >
> > > These three properties always hold for linear Gaussian additive noise SCMs for any number of parents (including the no parents scenario). We hope this clarifies your question about identifiability. Please let us know in case of more questions. Given we posted general comments with many additional experiments and also ran experiments to show that our results extend to the non-equal noise variance setting in response to your initial comment, "The model requires that all noise terms of latent variables have equal noise variance", it would be helpful if you could raise the score accordingly.
> > >
> > > [1] Brehmer, J., Haan, P. De, Lippe, P. & Cohen, T. Weakly supervised causal representation learning. in ICLR2022 Workshop on the Elements of Reasoning: Objects, Structure and Causality (2022).

---

> > > ### Author Response · Authors · 2022-12-11
> > > **Discussion ends in 24h; any further questions?**
> > >
> > > Dear reviewer Aq59,
> > >
> > > We believe we have addressed your concern about identifiability by showing that the causal mechanisms are invertible, differentiable, and the inverse mechanism is differentiable. Further, ref [1] also states that "any SCM can be brought to this form" which supports the identifiability results we rely on. In previous comments, we have also addressed your other concern on equal noise variance assumptions and performed experiments showing that the proposed method works even when this assumption is relaxed. We hope this answers your question. Please let us know if you have further questions and in case all of them are clarified, we request you to consider raising the score.

---

### Official Review · Reviewer_oCT2 · 2022-10-24

**Confidence:** 4
**Correctness:** 2
**Technical Novelty And Significance:** 2
**Empirical Novelty And Significance:** 3
**Recommendation:** 3

**Clarity, Quality, Novelty And Reproducibility:**

Clarity:

This paper is easy to follow.

Quality and Novelty:

The paper proposes a Bayesian-based estimation method for latent causal model estimation, which is novel to me. However, this paper does not give the theoretical identifiability of the latent causal model.

Reproducibility:
[Updates after rebuttal] Code is provided.

**Strength And Weaknesses:**

Strength:

1. The studied problem, learning latent causal models, is very important not only to representation learning itself, but also to other downstream tasks.

2. The authors estimate the latent causal model in a Bayesian way, which is different from previous estimation methods, as far as I know.

Weakness:

1. This paper lacks theoretical identifiability, which is essential when claiming causality.

2. For empirical estimation  (Figure 4), the MSE of the proposed method is always larger than other baselines when the causal ordering is not given. Can the authors give an explanation about it?

[Updates after rebuttal]: the authors showed that with more iterations, the performance of the proposed method gets better.

3. I would suggest the authors also compare with methods that do not put a prior on the latent causal structure and corresponding parameters.

**Summary Of The Paper:**

This paper considers the estimation of latent causal models, where low-level data like image pixels or high-dimensional vectors are observed, but not the underlying causal variables. Different from previous methods, the authors handle this problem in a Bayesian manner, by putting a prior on the latent causal structure and corresponding parameters.

**Summary Of The Review:**

The authors estimate the latent causal model in a Bayesian way, which is different from previous estimation methods. However,  this paper lacks theoretical identifiability, which is essential when claiming causality and is one of my main concerns. In addition, in empirical estimation (Figure 4), the MSE of the proposed method is always larger than other baselines, which may be problematic (does it mean that the proposed method is worse than others wrt MSE?)

---

> ### Author Response · Authors · 2022-11-12
> **Response to Reviewer oCT2 (1/3)**
>
> Thanks for your overall feedback and comments on our work!
>
> > 1. This paper lacks theoretical identifiability, which is essential when claiming causality.
>
> - We agree that questions of identifiability are important when making conclusions about the structure of a causal model, for methods returning only a single structure (”maximum likelihood methods”, as mentioned in our Related Work). However in our work, we have access to interventional data and approximate a full posterior distribution over the latent SCMs, instead of returning just a single graph. In this setting, questions of identifiability become less critical, as we can assign probabilities for many possible candidate graphs (and parameters) to express our level of confidence that a particular SCM yields the correct causal conclusions. This is a softer guarantee than what identifiability would provide. That being said, we would like to note that recent prior work on causal representation learning [1,2,3] has addressed these questions of identifiability under similar conditions as ours (identifying latent causal models from low-level observations), but again more applicable to “maximum-likelihood methods” returning a single graph.
>
> - Our setup is similar to [1] and in fact, our proposed method relies on the identifiability result of [1]. We have also acknowledged this fact in the results: **Section 5.1.2 paragraph on Results on nonlinear projection of causal variables**, “*This observation complements the identifiability result that recovery of latent variables is possible only upto a permutation in latent causal models [1,4] for general nonlinear mappings between causal variables and low-level data*”.
>
> - For completeness and to further aid the clarity of the paper, we have also **included a** **discussion on identifiability** in our revision in a new, separate section A.9 (”Discussion on identifiability”) of the appendix as well cited these works in the last paragraph of the related works section.
>
> [1] Brehmer, J., Haan, P. De, Lippe, P. & Cohen, T. Weakly supervised causal representation learning. in *ICLR2022 Workshop on the Elements of Reasoning: Objects, Structure and Causality* (2022).
>
> [2] Kartik Ahuja, Jason Hartford, & Yoshua Bengio. Weakly supervised representation learning with sparse perturbations.
>
> [3] Kartik Ahuja, Divyat Mahajan, Vasilis Syrgkanis, & Ioannis Mitliagkas. Towards efficient representation identification in supervised learning.
>
> [4] Yuhang Liu, Zhen Zhang, Dong Gong, Mingming Gong, Biwei Huang, Anton van den Hengel, Kun Zhang, and Javen Qinfeng Shi. Weight-variant latent causal models, 2022.

---

> > ### Comment · Reviewer_oCT2 · 2022-12-12
> > **Response to authors**
> >
> > Dear authors,
> >
> > Thank you very much for your detailed response. (I am very sorry for the late reply for some personal reasons.)
> >
> > Identifiability: Actually, I (as well as the field) think identifiability is necessary if claiming causality, no matter for frequency or Bayesian-based method. For Bayesian-based method that returns a list of causal graphs with different posterior probabilities, you need to show the true graph has the highest probabilities in theory with infinite data. For practical estimation with limited data, considering the top k graphs may be better than choosing the top one. For previous approaches to causal discovery with Bayesian methods, you may have a look at BDeu.
> >
> > Experimental results: Thanks for the update!

---

> > > ### Author Response · Authors · 2022-12-13
> > > **Response to reviewer oCT2**
> > >
> > > Dear reviewer oCT2,
> > >
> > > Thank you for your response. We believe you mentioned you have adjusted the score but this is not visible on openreview.
> > > We have also addressed identifiability concerns in a previous comment to other reviewers, please see [this comment](https://openreview.net/forum?id=w2mDq-p9EEf&noteId=ATM8IUmd7ve).
> > >
> > > - We think the main point of concern seems to be that it is hard to see how their identifiability results extend to our setting (because of differentiability and invertibility of causal mechanisms). We have also addressed this.
> > >
> > > Please refer page 3 of [Ref [1]](https://arxiv.org/abs/2203.16437) which states:
> > > > An SCM entails a unique solution s : $\varepsilon$ → $\mathcal{Z}$ defined by successively applying the causal mechanisms. We require the causal mechanisms to be pointwise diffeomorphic, that is, for any value of the [parents](https://arxiv.org/abs/2203.16437) $z_{pa_i}$ we have that $f_i(·; z_{pa_i})$ is invertible, differentiable, and its inverse is differentiable.
> > >
> > > The footnote at the end of the page includes the following statement which clarifies the concern:
> > > > Under some mild smoothness assumptions, any SCM can be brought into this form by elementwise redefinitions of the variables, preserving the observational and interventional distributions.
> > >
> > > - However, we also show that causal mechanisms are invertible and differentiable for linear Gaussian SCMs which is a sufficient condition for identifiability results of [1] to extend to our setting. We have shown this proof in [this comment](https://openreview.net/forum?id=w2mDq-p9EEf&noteId=ATM8IUmd7ve) and include the link here for brevity. We can promise to include a clearer version of this proof justifying identifiability in the camera-ready version of the paper, if accepted.
> > >
> > > - As for the BDeu work you have mentioned, it seems to be that the SCM is not learned from pixels? Could you perhaps include a citation so we can be sure we are looking at the right one?

---

> ### Author Response · Authors · 2022-11-12
> **Response to Reviewer oCT2 (2/3)**
>
> > 2. For empirical estimation (Figure 4), the MSE of the proposed method is always larger than other baselines. Can the authors give an explanation about it?
>
> Thanks for pointing out this interesting observation in our plots. Firstly, we would like to clarify that from the plots in figure 4, the MSE($L$, $\hat{L}$) of the proposed method is not *always* larger than other baselines. It is larger *only* in the experiment where we also learn the node ordering/permutation (2nd row, last column plot of figure 4).
>
> The MSE between ground truth and predicted edge weights is just a proxy metric for how well our method is performing and is *not a perfect metric*. Moreover, **this metric makes sense only for models that learn/infer the parameters** (or edge weights for linear SCMs), such as ours, and neither of the baselines have the ability to infer parameters. We however included the MSE between ground truth and predicted edge weights for the sake of completeness.
>
> Nevertheless, we speculated that the reason for higher MSE is that learning the permutation (aka node ordering) in the experiment (bottom row of Fig. 4), makes it a harder problem compared to other experiments. **We suspected that either more data or longer training would fix this issue**. To confirm this suspicion, we allowed the algorithm to run for more iterations and we see that MSE($L$, $\hat{L}$) do in fact get better with longer training and now performs better than both baselines ([figure 20](https://ibb.co/YTgH25s) of the appendix).
>
> To address yours and other reviewers’ comments (jaMU and Aq59), and also to remove some of our assumptions, we made the following changes to the experimental setup:
>
> - Moved from an equal-noise variance setting to non-equal noise variance setting which is more general.
> - Earlier, the observation generating function mapping from latent to observed variables was deterministic. We have now made this stochastic and also use a stochastic decoder now.
> - Train for longer with 10k iterations instead of the earlier 5k iterations.
>
> With all these changes done (all hyperparameters and other experimental conditions were kept constant), we re-ran the experiment for figure 4. We note that our method continues to perform best among baselines and that, as suspected, **longer training  does improve performance and causes the MSE($L$, $\hat{L}$) of the proposed method to outperform baselines**. We will replace figure 4 with this image soon in a future revision, but for now, we add the image to the end of the appendix (please see figure 20).
>
> Links for ease of access: [figure 4](https://ibb.co/0cPT6j3) (old), [figure 20](https://ibb.co/YTgH25s) (updated)

---

> ### Author Response · Authors · 2022-11-12
> **Response to Reviewer oCT2 (3/3)**
>
> > 3. I would suggest the authors also compare with methods that do not put a prior on the latent causal structure and corresponding parameters.
>
> - Thanks for the suggestion. We are not aware of any methods that solve this problem of learning a latent SCM from low-level observations (like pixels or high-dimensional vectors) — incorporating priors or otherwise — except a recent work [7]. However, the code for [7] has not been released yet. We were unable to obtain code even after correspondence with the authors. Since the rebuttal period is too short to reproduce their results reliably, we are unfortunately not able to compare to this work, as of now.
> - There are a few works that are closely related [1-6], as reviewer jaMU pointed out, but these are in a different setting where some variables of the causal model are observed and other variables are unobserved. Moreover, these works consider causal edges between the latent and observed variables.
> On the other hand, our setup involves a set of latent variables which explain some low-level high dimensional data, and a causal model is defined over the *entire set of latent variables*. There are *no causal edges between latent variables and the low-level data*.
> - Finally, while some of the above works provide relevant identifiability results (which have been cited in the revised version in the related work) and also algorithms for recovering structure/parameters for partially observed causal models, none of these algorithms are designed to operate on low-level data like image pixels or high-dimensional vectors.
>
> However, if you are aware of works that are in a similar setting, please let us know and we would be happy to add them as baselines.
>
> [1] Anandkumar, A., Hsu, D., Javanmard, A. & Kakade, S. Learning Linear Bayesian Networks with Latent Variables. in *Proceedings of the 30th International Conference on Machine Learning* (eds. Dasgupta, S. & McAllester, D.) vol. 28 249–257 (PMLR, 2013).
>
> [2] Xie, F. *et al.* Generalized Independent Noise Condition for Estimating Latent Variable Causal Graphs. in *Advances in Neural Information Processing Systems* (eds. Larochelle, H., Ranzato, M., Hadsell, R., Balcan, M. F. & Lin, H.) vol. 33 14891–14902 (Curran Associates, Inc., 2020).
>
> [3] Silva, R., Scheine, R., Glymour, C. & Spirtes, P. Learning the Structure of Linear Latent Variable Models. *J. Mach. Learn. Res.* **7**, 191–246 (2006).
>
> [4] Markham, A. & Grosse-Wentrup, M. Measurement dependence inducing latent causal models. in *Proceedings of the 36th Conference on Uncertainty in Artificial Intelligence, UAI 2020* (eds. Peters, J. & Sontag, D.) vol. 124 609–618 (PMLR, 2020).
>
> [5] Kivva, B., Rajendran, G., Ravikumar, P. & Aragam, B. Learning latent causal graphs via mixture oracles. in *Advances in Neural Information Processing Systems* (eds. Ranzato, M., Beygelzimer, A., Dauphin, Y., Liang, P. S. & Vaughan, J. W.) vol. 34 18087–18101 (Curran Associates, Inc., 2021).
>
> [6] Elidan, G., Lotner, N., Friedman, N. & Koller, D. Discovering Hidden Variables: A Structure-Based Approach. in *Advances in Neural Information Processing Systems* (eds. Leen, T., Dietterich, T. & Tresp, V.) vol. 13 (MIT Press, 2000).
>
> [7] Brehmer, J., Haan, P. De, Lippe, P. & Cohen, T. Weakly supervised causal representation learning. in *ICLR2022 Workshop on the Elements of Reasoning: Objects, Structure and Causality* (2022).
>
> > Reproducibility: Code is not provided, as well as details of the implementation.
>
> **We would like to note that our initial submission already includes code** (please see the [anonymized link](https://anonymous.4open.science/r/anon-biols-86E7) to the code in our Reproducibility statement). Please let us know in case you are not able to access the code. To address your concerns regarding implementation details, we have added a section “Implementation details” for this in the appendix (section A.2) of the revised paper.

---

> ### Author Response · Authors · 2022-11-24
> **Further questions to be addressed?**
>
> Dear Reviewer oCT2,
>
> Thank you again for your review. Please let us know if you have further questions so we can continue the discussion and see if there are any issues left. If there are no further questions, we would be happy if you could consider raising the score.

---

> ### Author Response · Authors · 2022-12-07
> **Discussion period coming to an end; any remaining concerns?**
>
> Dear Reviewer oCT2,
>
> This is a gentle reminder that the discussion period is coming to an end. We have addressed all concerns brought up in our detailed responses and we would like to know if there are any remaining concerns? If there are no remaining concerns, adjusting the rating accordingly would be helpful.

---

> ### Author Response · Authors · 2022-12-11
> **Discussion period ends in ~24h; further questions to be addressed?**
>
> Dear Reviewer oCT2,
>
> This is a gentle reminder that the discussion period is coming to an end. We have addressed all concerns brought up in our detailed responses and we would like to know if there are any remaining concerns?
>
> - Theoretical identifiability: we have addressed this in comments, included a section discussing identifiability and in short identifiability has been studied in other works [1] which we rely on. Our contribution is not the theoretical identifiability, but rather on proposing a model that can learn from low-level data. However, we also have discussed why identifiability results of [1] hold in our case in a response to reviewer Aq59 (please see [this comment](https://openreview.net/forum?id=w2mDq-p9EEf&noteId=ATM8IUmd7ve)).
>
> - "the MSE of the proposed method is always larger than other baselines. Can the authors give an explanation about it": we have also answered this in our [earlier response](https://openreview.net/forum?id=w2mDq-p9EEf&noteId=W1K3bIPHMls) to your comment. Apart from providing a reasonable explanation, we have also validated this hypothesis with an experiment and improved our results and now the MSE of the proposed method is better than baselines. This can be seen in figure 20 in the latest version of the paper.
>
> - Finally, we have also addressed your comment on "compare with methods that do not put a prior on the latent causal structure" in [response 3/3](https://openreview.net/forum?id=w2mDq-p9EEf&noteId=XJEmrgEWHs) in this thread.
>
> If there are no remaining concerns, we kindly request you to consider the raising the score.
>
> [1] Brehmer, J., Haan, P. De, Lippe, P. & Cohen, T. Weakly supervised causal representation learning. in ICLR2022 Workshop on the Elements of Reasoning: Objects, Structure and Causality (2022).

---

### Official Review · Reviewer_pPfL · 2022-10-24

**Confidence:** 4
**Correctness:** 3
**Technical Novelty And Significance:** 3
**Empirical Novelty And Significance:** 3
**Recommendation:** 8

**Clarity, Quality, Novelty And Reproducibility:**

Overall the submission is very clearly written, is of high quality, and appears to be novel and reproducible. Overall, a very nice submission!

That being said, it would be helpful to further emphasize which elements of the work are an assembly and synthesis of existing techniques (which is itself valuable!) and which are truly novel.

**Strength And Weaknesses:**

This submission has many strengths, and few weaknesses. Overall, I believe this submission would make a strong contribution to the ICLR conference.

Strengths:
The authors do an excellent job of communicating the problem statement, the problem's technical challenges (simultaneous optimization over representations and structure), and their solution (factorize the representation and structure and optimize via variational methods).  The proposed approach is intuitive, yet involves a number of technical details to make the approach tractable. For example, the use of continuous relaxations, while not novel to this work, is a nice detail.

Opportunities for improvement:
My main concern with this submission is the implicit claim that being Bayesian about structure learning implies that we no longer need to think about identifiability w.r.t causal graph structure. In the Bayesian setting, non-identifiability means that the posterior will not converge to a single maximum likelihood solution (i.e. assumptions for Bernstein von Mises are violated). In other words, the posterior distribution will always maintain some uncertainty over structures that are likelihood equivalent. While this would not be a concern if the inference method were exact (or even asymptotically consistent), that is not the case for variational inference. All this is to say that relying on variational inference to identify non-identifiability (in this case likelihood equivalence amongst graphs) seems risky. Now, it's possible that identifiability is not a concern here because of the use of interventional data. If so, that would be an excellent point to clarify and discuss in a revision.

Besides that concern, it would generally be helpful to clarify a bit more about what it means to intervene on latent variables, especially those that don't a-priori map to recognizable concepts in the domain. In other words, what does it mean to have a known intervention on unknown variables?

Finally, the empirical study would be improved by quantifying the performance of the learned graph on some downstream task directly. That authors explore this with out of sample prediction, but it would be helpful to expand on these a bit more.

**Summary Of The Paper:**

This submission presents a a variational Bayesian approach to causal structure learning on learned representations of high-dimensional data. Specifically, this paper considers the problem where there are random, known interventions. The authors present a custom ELBO for this structure learning task, and use it optimize a variational approximation over graph matrices and model parameters conditional on data. Finally, the authors present empirical results that demonstrate state-of-the-art performance on synthetic data generating processes and on real applications on image data.

**Summary Of The Review:**

Overall this submission is well written, and clearly articulates arguments and evidence in support of its key claims.

---

> ### Author Response · Authors · 2022-11-18
> **Response to reviewer pPfL**
>
> > This submission has many strengths, and few weaknesses. Overall, I believe this submission would make a strong contribution to the ICLR conference.
> > Strengths: The authors do an excellent job of communicating the problem statement, the problem's technical challenges (simultaneous optimization over representations and structure), and their solution (factorize the representation and structure and optimize via variational methods). The proposed approach is intuitive, yet involves a number of technical details to make the approach tractable. For example, the use of continuous relaxations, while not novel to this work, is a nice detail.
> >
> We thank the reviewer for the overall positive response and also for their recommendations to improve our manuscript.
>
> > Opportunities for improvement: My main concern with this submission is the implicit claim that being Bayesian about structure learning implies that we no longer need to think about identifiability w.r.t causal graph structure. In the Bayesian setting, non-identifiability means that the posterior will not converge to a single maximum likelihood solution (i.e. assumptions for Bernstein von Mises are violated). In other words, the posterior distribution will always maintain some uncertainty over structures that are likelihood equivalent. While this would not be a concern if the inference method were exact (or even asymptotically consistent), that is not the case for variational inference. All this is to say that relying on variational inference to identify non-identifiability (in this case likelihood equivalence amongst graphs) seems risky. Now, it's possible that identifiability is not a concern here because of the use of interventional data. If so, that would be an excellent point to clarify and discuss in a revision.
>
>
> - We would like to thank the reviewer for this comment. As noted, we argue in this submission that problems of identifiability become less critical when we have a distribution over SCMs. However the reviewer is correct in saying that since we are using variational inference, guarantees about the (true) posterior distribution overs SCMs may not transfer to the posterior approximation we use. We conjecture that our approximation of the posterior distribution with the variational distribution $q_{\phi}$ is flexible enough to provide an accurate approximation of $P(\mathcal{Z}, \mathcal{G}, \Theta \mid \mathcal{D})$.
> - Our setup is similar to [1] and in fact, our proposed method relies on the identifiability result of [1]. We have also acknowledged this fact in the results: **Section 5.1.2 paragraph on Results on nonlinear projection of causal variables**, “*This observation complements the identifiability result that recovery of latent variables is possible only upto a permutation in latent causal models [1,4] for general nonlinear mappings between causal variables and low-level data*”.
> - We have also **included a** **discussion on identifiability** in our revision in a new, separate section A.9 (”Discussion on identifiability”) of the appendix as well cited these works [1-4] in the last paragraph of the related works section.
>
> We have also added additional content in the manuscript and extended our experiments to stochastic observation generating process, and non-equal noise variance setting. We have made a summary of the main changes in [this comment](https://openreview.net/forum?id=w2mDq-p9EEf&noteId=2QLpOYxZgmR). We are happy to address further questions, if any.
>
> [1] Brehmer, J., Haan, P. De, Lippe, P. & Cohen, T. Weakly supervised causal representation learning. in *ICLR2022 Workshop on the Elements of Reasoning: Objects, Structure and Causality* (2022).
>
> [2] Kartik Ahuja, Jason Hartford, & Yoshua Bengio. Weakly supervised representation learning with sparse perturbations.
>
> [3] Kartik Ahuja, Divyat Mahajan, Vasilis Syrgkanis, & Ioannis Mitliagkas. Towards efficient representation identification in supervised learning.
>
> [4] Yuhang Liu, Zhen Zhang, Dong Gong, Mingming Gong, Biwei Huang, Anton van den Hengel, Kun Zhang, and Javen Qinfeng Shi. Weight-variant latent causal models, 2022.

---

> ### Author Response · Authors · 2022-12-07
> **Discussion period coming to an end; any remaining concerns?**
>
> Dear Reviewer pPfL,
>
> This is a gentle reminder that the discussion period is coming to an end. We have addressed all concerns brought up in our detailed responses and we would like to know if there are any remaining concerns?

---

### Official Review · Reviewer_jaMU · 2022-10-24

**Confidence:** 4
**Correctness:** 3
**Technical Novelty And Significance:** 2
**Empirical Novelty And Significance:** 2
**Recommendation:** 3

**Clarity, Quality, Novelty And Reproducibility:**

Can you provide an evaluation of the quality, clarity and originality of the work?

- Overall, writing quality and clarity should be improved:
  - I am especially unhappy about the confusion between random variables and their realizations (e.g., the definition in Sec. 4.1, the paragraph below Eq. (7), in the paragraph "Evaluation Metrics", ...). This makes the paper harder to read, e.g. when determining whether p(Z | ..) is a conditional distribution/density or its evaluation, and is simply sloppy.
  - I would have wished for a more thorough and consistent motivation in the introduction of the problem scenario presented.

- Regarding originality, I miss the discussion of earlier, closely related work and do not agree with the claim that the authors are "the first to study this task of learning latent SCMs from low level observations..." (see my comments above).
- From a methodological point of view the authors derive a Variational Inference objective (with issues; see above), and extend the causal discovery method from the recently proposed BCD Nets paper by estimating the observation generation conditional $q_\psi$  and incorporating the use of interventional data (cf. Algs. 1 from the present work and the BCD Nets paper).  Thus, the contribution from a methods perspective is only minor.
- In Section 5.2 it is not clear whether the causal ordering is given or learned. The generalization demonstration in Fig. 7 is not really conclusive; a quantitative evaluation would be desirable (also comparing to baselines).
- Regarding reproducibility, the authors apparently enclosed the code for producing the experiments, I did however not review it.

**Strength And Weaknesses:**

The idea of employing a Bayesian approach to the latent structure learning problem is certainly an interesting and promising one. However, the form and content of the paper leave many things to be desired.

- The main weakness of this work stems from the rigid assumptions made about the problem setup, i.e.,
  - the authors assume the *number of latent variables*, and the *latent intervention targets* for the corresponding observed samples to be known. From an empirical perspective, I find these assumptions rather problematic: assuming there exists some ground truth latent data generating process it is not clear how one can reliably determine the true number of latent variables involved, and relying on expert knowledge seems very brittle (e.g., consider an image generating process; what are the actual latent variables: color, lighting, what else?). In case there is a mismatch between the true and the assumed number of latents, the true latent SCM cannot be identified anymore. It is not discussed, whether or not this work can obtain a causally consistent (cf. [8,9] ) latent SCM in this case, or more interestingly, whether the posterior over latent SCMs converges to the set of causally consistent latent SCMs.
    Additionally assuming the latent intervention targets to be known seems like a long stretch and untestable in practice. Do the authors have a practical application scenario in mind where this could be reasonably assumed?
  - Additionally, the authors assume the *causal order of the latent variables* in parts of their experiments to be known. This simplifies the structure learning problem greatly but is generally not known. In the experiments, the proposed method (understandably) performs significantly worse when the causal order is learned as well. However, in the non-linear projection case performance deteriorates so much as to the performance of predicting a null graph, which greatly limits applicability in practice.
  - The assumption of Gaussian noise and equal noise variance in the latent SCM, let alone homoscedasticity, are convenient tractability assumptions, but rather restrictive for many application scenarios. As far as I can see, the objective in (7) relies on the assumption that the variational $q_\phi(Z | G, \theta)$ and its corresponding true density have the same form and only depend on the parameters $\theta$, so it does not in general hold for the non-Gaussian setting. This is a minor restriction in light of the other assumptions though.
  - the authors assume, that the observation generating process p(**X** | **z**) is a deterministic projection function. This is also a very strong assumption: essentially any kind of physical measurement (e.g., measuring the physical position of a robots' arm, taking a photograph with your smartphone, etc.) will be afflicted with noise.

​		A concise summary of these assumptions (maybe in tabular form comparing with related work) may greatly aid the overall clarity of the paper.

- Missing discussion of related work/baselines:
  Recent and older important related work about latent structure learning and identifiability (in linear models) that was unfortunately not considered, and includes for instance (non-exhaustive) [1-6]. These works establish various identifiability criteria for feasibly latent structure learning, and in part even consider the more realistic and harder scenario where the *number of latents* is unknown, and/or only observational data is available, and/or more complex between observed and latent variables are considered. Although these proposed methods are not Bayesian, an experimental comparison, at least to some of these works, would be feasible and necessary for a serious evaluation.
- In line with the above, the authors could have compared their work with [7], which assumes that the occurrence of interventions is known but that their targets are unknown. This work, by assuming known intervention targets makes even stronger assumptions. Furthermore, the experimental setup is rather unfair w.r.t. the baseline methods: VAEs assume the latents to be independent and GraphVAE assumes all edge weights to be 1. Both are scenarios that could have been explored easily by constructing corresponding ground truth models and see how well the proposed method performs in this case.
- Missing discussion of identifiability:
  Latent structure learning is a hard and in general infeasible endeavor. The strict assumptions made by the authors may help in that regard, but are not discussed w.r.t. identifiability. The suggested remedy of adopting a Bayesian problem formulation (cf. Introduction) is not too helpful in that regard, for instance consider [3, Sec. 3]: "the equivalence class of all latent variable models that cannot be distinguished given the likelihood function might be very large. [...] A representation of such an equivalence class [...] can be cumbersome and uninformative".  An according discussion is thus crucial in my opinion.
- Considering the issues above, I miss a reasonable discussion of limitations of the proposed work.

- Correctness:
  - In the paragraph "Alternate factorization of the posterior" the authors write "Thus, the prior $p(Z | G, \Theta)$ and the posterior $p(Z | G, \Theta, D) = q_\phi(Z | G, \Theta)$  are identical." I disagree here. According to the BN in Fig. 3 the true posterior is not conditionally independent from $D$ given $G,\Theta$ and reads
    $$p(Z | G, \Theta, D) = \frac{p(D | Z)}{p(D | G, \Theta)} \cdot p(Z | G, \Theta)$$, which is not the same.
  -  As far as I can see, the simplification in Eq. (7) only holds, because the variational $q_\phi(Z | G, \theta)$ and its corresponding true density have the same form and are fully specified by the parameters $\theta$ .
  - The proof in Appendix A makes this assumption right away in the first line, without explanation. Furthermore, in the derivation the parameters of the observation generating density $q_\psi(D | Z)$ are not considered and the derived objective contains the *true* observation generating density $p(D | Z)$. In Eq. (7) this is simply assumed to be the variational $q_\psi$ which is not valid in general. The parameters $\psi$ should be included in the VI objective.

[1] Anandkumar, A., Hsu, D., Javanmard, A. & Kakade, S. Learning Linear Bayesian Networks with Latent Variables. in *Proceedings of the 30th International Conference on Machine Learning* (eds. Dasgupta, S. & McAllester, D.) vol. 28 249–257 (PMLR, 2013).

[2] Xie, F. *et al.* Generalized Independent Noise Condition for Estimating Latent Variable Causal Graphs. in *Advances in Neural Information Processing Systems* (eds. Larochelle, H., Ranzato, M., Hadsell, R., Balcan, M. F. & Lin, H.) vol. 33 14891–14902 (Curran Associates, Inc., 2020).

[3] Silva, R., Scheine, R., Glymour, C. & Spirtes, P. Learning the Structure of Linear Latent Variable Models. *J. Mach. Learn. Res.* **7**, 191–246 (2006).

[4] Markham, A. & Grosse-Wentrup, M. Measurement dependence inducing latent causal models. in *Proceedings of the 36th Conference on Uncertainty in Artificial Intelligence, UAI 2020* (eds. Peters, J. & Sontag, D.) vol. 124 609–618 (PMLR, 2020).

[5] Kivva, B., Rajendran, G., Ravikumar, P. & Aragam, B. Learning latent causal graphs via mixture oracles. in *Advances in Neural Information Processing Systems* (eds. Ranzato, M., Beygelzimer, A., Dauphin, Y., Liang, P. S. & Vaughan, J. W.) vol. 34 18087–18101 (Curran Associates, Inc., 2021).

[6] Elidan, G., Lotner, N., Friedman, N. & Koller, D. Discovering Hidden Variables: A Structure-Based Approach. in *Advances in Neural Information Processing Systems* (eds. Leen, T., Dietterich, T. & Tresp, V.) vol. 13 (MIT Press, 2000).

[7] Brehmer, J., Haan, P. De, Lippe, P. & Cohen, T. Weakly supervised causal representation learning. in *ICLR2022 Workshop on the Elements of Reasoning: Objects, Structure and Causality* (2022).

[8] Rubenstein, P. K. *et al.* Causal Consistency of Structural Equation Models. in *Proceedings of the 33rd Conference on Uncertainty in Artificial Intelligence (UAI)* (Association for Uncertainty in Artificial Intelligence (AUAI), 2017).

[9] Bongers, S., Forré, P., Peters, J. & Mooij, J. M. Foundations of structural causal models with cycles and latent variables. *Ann. Stat.* **49**, (2021).

**Summary Of The Paper:**

This paper proposes a variational inference approach to infer a Structural Causal Models (SCM) over latent variables given interventional data collected from a higher-dimensional observation space under the following assumptions: the latent SCM is linear with equal-variance Gaussian additive noise, the number of latent variables is known, and the observation generating function mapping from latent to observed variables is deterministic. They evaluate their method experimentally and compare to VAE and GraphVAE models.

**Summary Of The Review:**

A substantial revision of the presented work is necessary due to the following issues:

- An essential discussion of identifiability of the true latent SCM, and/or causal consistency of the learned SCM is missing; without it, making causal claims is contestable and the proposed method amounts to fitting a distribution over observed variables with a hierarchical latent space model.
- The submitted work lacks substantially in the discussion of and experimental comparison to related work.
- Issues of correctness/preciseness in the derived utility need to be addressed.
- The strong assumptions made should be concisely presented, compared to related work, and at least well justified.
- The methodological contributions are rather shallow.

---

> ### Author Response · Authors · 2022-11-15
> **Response to Reviewer jaMU (1/n)**
>
> Thanks a lot for your insightful comments and feedback. Over the next 1-2 days we will comment to address your concerns one by one.
>
> > In line with the above, the authors could have compared their work with [7], which assumes that the occurrence of interventions is known but that their targets are unknown. This work, by assuming known intervention targets makes even stronger assumptions.
>
> [7] looks like a good baseline to compare with but the code for the same has not been released yet. We were unable to obtain code even after correspondence with the authors. Since the rebuttal period is too short to reproduce their results reliably, we are unfortunately not able to compare to this work, as of now.
>
> [7] Brehmer, J., Haan, P. De, Lippe, P. & Cohen, T. Weakly supervised causal representation learning. in *ICLR2022 Workshop on the Elements of Reasoning: Objects, Structure and Causality* (2022).
>
> > Furthermore, the experimental setup is rather unfair w.r.t. the baseline methods: VAEs assume the latents to be independent and GraphVAE assumes all edge weights to be 1. Both are scenarios that could have been explored easily by constructing corresponding ground truth models and see how well the proposed method performs in this case.
>
> VAE and GraphVAE are not the ideal baselines but we compared with these since to the best of our knowledge, no other methods/baselines exist to learn latent SCMs from low-level observations. This is also stated in [7], where the baselines are VAE and slot-attention. "Both are scenarios that could have been explored easily by constructing corresponding ground truth models" -- this would basically mean that we test on GT DAGs that do not have edges (to be fair to VAEs) or test on GT DAGs where all edges have a weight of 1 (to be fair to GraphVAE). In the first case inferring a structure would not even be necessary and in the second case inference over parameters would not be necessary. Inferring the structure and parameters of a latent SCM is the focus of this work and having such assumptions on the GT DAG/parameters would be restrictive. However, please let us know if you still feel this is an important experiment to be included and we would be glad to do so.
>
> > Considering the issues above, I miss a reasonable discussion of limitations of the proposed work.
>
> We have concisely included and acknowledged all our limitations under a new section 6, in our latest revision.
>
> > - In the paragraph "Alternate factorization of the posterior" the authors write "Thus, the prior $p(Z \mid G, \Theta)$ and the posterior $p(Z \mid G, \Theta, D)=q_{\phi}(Z \mid G, \Theta)$ are identical." I disagree here. According to the BN in Fig. 3 the true posterior is not conditionally independent from $D$ given $G, \Theta$ and reads $\begin{equation} p(Z \mid G, \Theta, D) = \frac{p(D \mid Z)}{p(D \mid G, \Theta)} \cdot p(Z \mid G, \Theta)\end{equation}$, which is not the same.
> > - As far as I can see, the simplification in Eq. (7) only holds, because the variational $q_{\phi}(Z \mid G, \theta)$ and its corresponding true density have the same form and are fully specified by the parameters $\theta$.
>
> Thanks for pointing this out! We agree that $p(Z \mid G, \theta, \mathcal{D}) \neq q_{\phi}(Z \mid G, \theta)$. We have revised the text accordingly in the paragraph "Alternate factorization of the posterior": Rather than factorizing as in equation 6, we propose to only introduce a variational distribution $q_{\phi}(\mathcal{G}, \Theta)$ over structures and parameters, so that the approximation is given by $q_{\phi}(\mathbf{Z}, \mathcal{G}, \Theta) = p(\mathbf{Z}\mid \mathcal{G}, \Theta) \cdot q_{\phi}(\mathcal{G}, \Theta)$. The advantage of this factorization is that the distribution $p(\mathbf{Z}\mid \mathcal{G}, \Theta)$ over $\mathbf{Z}$ is completely determined from the SCM given $(\mathcal{G}, \Theta)$ and exogenous noise variables (assumed to be Gaussian). This conveniently avoids the hard simultaneous optimization problem mentioned in the text since optimizing for $q_\phi(\mathbf{Z})$ is not necessary.
>
> > - The proof in Appendix A makes this assumption right away in the first line, without explanation. Furthermore, in the derivation the parameters of the observation generating density $q_{\psi}(D \mid Z)$ are not considered and the derived objective contains the *true* observation generating density $p(D \mid Z)$. In Eq. (7) this is simply assumed to be the variational $q_{\psi}$ which is not valid in general. The parameters $\psi$ should be included in the VI objective.
>
> We have added the explanation to the Appendix A instead of making the assumption. Thanks for pointing out the missing parameters $\psi$, our intention was not to refer to the *true* observation generating density. We have modified the derivation to mention $p_{\psi}(\mathcal{D} \mid Z)$ instead. The parameters $\psi$ have also been included in the VI objective.

---

> ### Author Response · Authors · 2022-11-15
> **Response to reviewer jaMU (2/n)**
>
> > - The assumption of Gaussian noise and equal noise variance in the latent SCM, let alone homoscedasticity, are convenient tractability assumptions, but rather restrictive for many application scenarios. As far as I can see, the objective in (7) relies on the assumption that the variational $q_{\phi}(Z \mid G, \theta)$ and its corresponding true density have the same form and only depend on the parameters $\theta$, so it does not in general hold for the non-Gaussian setting. This is a minor restriction in light of the other assumptions though.
> > - The authors assume, that the observation generating process $p(\mathbf{X} | \mathbf{z})$ is a deterministic projection function. This is also a very strong assumption: essentially any kind of physical measurement (e.g., measuring the physical position of a robots' arm, taking a photograph with your smartphone, etc.) will be afflicted with noise.
>
> **Regarding the assumptions of equal noise variance and deterministic observation generating process** $p(\mathbf{X} \mid \mathbf{z})$, we performed experiments on linear and nonlinear projection of vector data (corresponding to [figure 4](https://ibb.co/0cPT6j3) and [figure 5](https://ibb.co/SxsGvVH) of the paper) after removing both these assumptions. Particularly, we ran experiments with non-equal noise variance and stochastic GT observation generating process. We also made our decoder stochastic. We ran these experiments for 10000 iterations (compared to the earlier 5000) to ensure convergence. **We summarize the results after removing these assumptions in [figure 20](https://ibb.co/YTgH25s) and [figure 21](https://ibb.co/VBpmS1M) respectively**. These correspond to experiments in figure 4 and 5 but without the assumptions.  We observe that **the proposed method continues to outperform baselines and recover the latent SCM even after these assumptions have been relaxed.**
>
> > - Authors assume the number of latent variables, and the latent intervention targets for the corresponding observed samples to be known. From an empirical perspective, I find these assumptions rather problematic: assuming there exists some ground truth latent data generating process it is not clear how one can reliably determine the true number of latent variables involved, and relying on expert knowledge seems very brittle (e.g., consider an image generating process; what are the actual latent variables: color, lighting, what else?).
>
> **Assumption on number of latent variables:** It is true that we assume the number of latent variables to be known. But we would like to point out that **this is a common assumption made in the literature** -- all the baselines (VAE and GraphVAE) and all prior works in causal representation learning have the same assumption (e.g., [1-3]). More research is required to address the case where the number of latent variables is not known apriori, as this would mean that the Bayesian causal discovery module needs to be able to deal with predicting the number of causal variables, and is a significantly harder problem. We believe that treatment of such a problem merits a separate work on its own and is beyond the current scope of this paper as well as most current causal representation learning methods.
>
> **Assumption of known intervention targets**: We noticed that assuming access to such an intervention improved our performance significantly. Though the identifiability from [1] requires only unknown interventions, identifiability does not necessarily imply learnability, and our need for known intervention targets for improved performance is a result of the fact that we learn a full distribution over the causal models. That being said, relaxing this assumption and extending to a setting that infers intervention targets (as in [4]) is an interesting avenue for future work.
>
> We have noted both the above assumptions under a new section 6 on limitations in our latest revision. Please let us know if you have further concerns and we would be happy to address them.
>
> [1] Brehmer, J., Haan, P. De, Lippe, P. & Cohen, T. Weakly supervised causal representation learning (2022).
>
> [2] Mengyue Yang, Furui Liu, Zhitang Chen, Xinwei Shen, Jianye Hao, and Jun Wang. Causalvae: Disentangled representation learning via neural structural causal models (2021).
>
> [3] Xinwei Shen, Furui Liu, Hanze Dong, Qing Lian, Zhitang Chen, and Tong Zhang. Disentangled generative causal representation learning (2020)
>
> [4] Alexander H ̈agele, Jonas Rothfuss, Lars Lorch, Vignesh Ram Somnath, Bernhard Sch ̈olkopf, and Andreas Krause. Bacadi: Bayesian causal discovery with unknown interventions.

---

> ### Author Response · Authors · 2022-11-15
> **Response to reviewer jaMU (3/n)**
>
> > A concise summary of these assumptions (maybe in tabular form comparing with related work) may greatly aid the overall clarity of the paper:
>  > - Missing discussion of related work/baselines: Recent and older important related work about latent structure learning and identifiability (in linear models) that was unfortunately not considered, and includes for instance (non-exhaustive) [1-6]. These works establish various identifiability criteria for feasibly latent structure learning, and in part even consider the more realistic and harder scenario where the number of latents is unknown, and/or only observational data is available, and/or more complex between observed and latent variables are considered. Although these proposed methods are not Bayesian, an experimental comparison, at least to some of these works, would be feasible and necessary for a serious evaluation.
>
> We have now included a short discussion of these papers [1-7] under the paragraph on "Structure learning with latent variables" in Related Works (section 3). We will include a slightly more detailed list of assumptions in these works and their settings. For finer details, one can view all assumptions, approach summary, and problem setting of these works [1-7] at [this link](https://i.imgur.com/SdF1no0.png). We kindly request reviewer(s) to have a look at this image which enumerates all the assumptions in these works.
>
> Comments general to works [1-6]:
> - All/most of these works [1,2,3,5,6] are in the setting of a **partially observed causal model or latent confounders**. Some variables/nodes are observed while others are unobserved. The observed nodes are assumed to be effects and cannot cause (observed or unobserved variables). In contrast, ours is in a setting where the entire SCM is latent. All variables, their structure, and SCM parameters have to be learnt.
> - Some of these works [5,6] are also **in the setting of discrete latent variables**. In constrast, ours in a setting where the latent variables are continuous. Some of these works are in a setting that assumes non-Gaussian noise [1,2]. In contrast, ours is in a Gaussian noise setting.
> - [4] does not use the concept of SCM but defines a new notion of causal models called MCM. Moreover, this work **does not allow for causal links between latent variables** (which is a key focus of our work). The binary adjacency matrix that is estimated corresponds to connections between latent variables and observed variables. In contrast, the structure that we want to recover in this work is the structure among latent variables.
> - Some of these works also place different **assumptions on the structure of the adjacency matrix** [1,2,5]. In contrast, we do not make any assumptions on the GT adjacency matrix.
> - Many of these works [1,2,3] are also in a setting where the causal model is linear. While this is also our assumption, the entire latent SCM being latent still allows for a nonlinear relationship between the latent causal variables and the observed data. On the other hand, in [1,2,3], the observed variables are restricted to linear combinations of causal variables. Thus, **we study a more complex relationship between latent causal variables and observed data**.
> - Finally, **none of the algorithms proposed in [1-6] are for handling low-level data** like high-dimensional vectors or images pixels (though the motivation of one these works mentions image pixels, but the algorithm is not built to handle images).
> - Since our SCM is latent, our search space is over $d$-node DAGs. However, these works often consider the observed and unobserved variables to be part of the DAG which has $D+d$ nodes. For representation learning scenarios, we have $D >> d$. Since the space of DAGs grows super-exponentially in the number of nodes, the setting in these works might involve searching a larger space of DAGs as compared to ours.
> - Many of these approaches seem to not discuss about interventions, which we consider in our work.
>
> [1] Anandkumar, A., Hsu, D., Javanmard, A. & Kakade, S. Learning Linear Bayesian Networks with Latent Variables (PMLR, 2013).
>  Comment
>
> [2] Xie, F. et al. Generalized Independent Noise Condition for Estimating Latent Variable Causal Graphs (NeurIPS, 2020)
>
> [3] Silva, R., Scheine, R., Glymour, C. & Spirtes, P. Learning the Structure of Linear Latent Variable Models (2006).
>
> [4] Markham, A. & Grosse-Wentrup, M. Measurement dependence inducing latent causal models (2020).
>
> [5] Kivva, B., Rajendran, G., Ravikumar, P. & Aragam, B. Learning latent causal graphs via mixture oracles (2021).
>
> [6] Elidan, G., Lotner, N., Friedman, N. & Koller, D. Discovering Hidden Variables: A Structure-Based Approach (MIT Press, 2000).
>
> [7] Brehmer, J., Haan, P. De, Lippe, P. & Cohen, T. Weakly supervised causal representation learning (2022).

---

> ### Author Response · Authors · 2022-11-15
> **Response to reviewer jaMU (4/n)**
>
> Apart from the general assumptions/comments given in our previous response (3/n), we will now concisely try to summarize the assumptions specific to each of these works.
>
> **Learning Linear Bayesian Networks with Latent Variables** [1]:
> - Hidden variables are assumed to be linearly independent.
> - Rank condition: There exists a fixed partition $\mathcal{P}$ of $[n]$ such that $|\mathcal{P}|=3$ and submatrix $A_I$ has full column matrix for all $I \in \mathcal{P}$ ($A$ is the adjacency matrix).
> - Graph expansion property of $A$ is assumed: every subset of hidden variables has "enough" outgoing edges.
> - Adjacency matrix has the parameter genericity property (imposes a constraint on the DAG structure).
> - Implementing the algorithm requires an incoherence assumption on $A$ to find the partition $\mathcal{P}$ (section 3.4).
>
> **Generalized Independent Noise Condition for Estimating Latent Variable Causal Graphs** [2]:
> - Observed variables are a linear combination of parent hidden variables.
> - Noise terms are non-Gaussian.
> - Double-Pure child assumption (places a constraint on GT DAG structure): Each latent variable set $L'$, in which every latent variable directly causes the same set of observed variables, has at least $2Dim(L')$ pure measurement variables as children.
>
> **Measurement dependence inducing latent causal models** [4]:
> - Does not allow for causal links between latent variables, thus there is no structure in the latent space, just connections between the latent and observed variables.
> - Constraint based method and the approach is limited by the accuracy of performing nonlinear conditional independence tests.
> - The work does not discuss interventional data.
>
>
> **Learning latent causal graphs via mixture oracles** [5]:
> - Discrete latent variable assumption.
> - No twins (assumption 2.2, pg 6): Each hidden variable has a unique set of neighbours.
> - Maximality (assumption 2.3, pg 6)
> - Non-degeneracy (assumption 2.4, pg, 6)
> - Subset condition (assumption 3.1, pg 8): For all distinct pairs of hidden variables $H_i, H_j$, the observed neighbors of $H_i$ cannot be a subset of the observed neighbors of $H_j$ (places a constraint on GT DAG structure).
> - Efficiently obtaining the adjacency matrix (Theorem 3.2) requires the columns of adjacency matrix to be linearly independent (places a constraint on GT DAG structure).
>
> **Discovering Hidden Variables: A Structure-Based Approach** [6]:
> - Discrete latent variable assumption.
> - The main assumption is that one can find structural signatures of hidden variables via semi-cliques.
> - Hidden variables in experiments are selected to be those variables that have a lot of observable children.
>
> **Weakly supervised causal representation learning** [7]:
> - Assumes access to pairs of observational and interventional data.
> - The algorithm has a label to determine if each data point is interventional or not.
> - Number of causal variables are known.
> - Identifiability results are only upto a permutation.
>
> **Learning Latent Structural Causal Models** (ours):
> - Gaussian noise assumption.
> - Intervention targets are known.
> - Number of causal variables are known.
> - Causal ordering (permutation) for part of the experiments are known.

---

> ### Author Response · Authors · 2022-11-16
> **Response to reviewer jaMU (5/n)**
>
> > Missing discussion of identifiability: Latent structure learning is a hard and in general infeasible endeavor. The strict assumptions made by the authors may help in that regard, but are not discussed w.r.t. identifiability. The suggested remedy of adopting a Bayesian problem formulation (cf. Introduction) is not too helpful in that regard, for instance consider [3, Sec. 3]: "the equivalence class of all latent variable models that cannot be distinguished given the likelihood function might be very large. [...] A representation of such an equivalence class [...] can be cumbersome and uninformative". An according discussion is thus crucial in my opinion.
>
> - We agree that questions of identifiability are important when making conclusions about the structure of a causal model, for methods returning only a single structure (”maximum likelihood methods”, as mentioned in our Related Work). However in our work, we have access to interventional data and approximate a full posterior distribution over the latent SCMs, instead of returning just a single graph. In this setting, questions of identifiability become less critical, as we can assign probabilities for many possible candidate graphs (and parameters) to express our level of confidence that a particular SCM yields the correct causal conclusions. This is a softer guarantee than what identifiability would provide. That being said, we would like to note that recent prior work on causal representation learning [1,2,3] has addressed these questions of identifiability under similar conditions as ours (identifying latent causal models from low-level observations), but again more applicable to “maximum-likelihood methods” returning a single graph.
> - Our setup is similar to [1] and in fact, our proposed method relies on the identifiability result of [1]. We have also acknowledged this fact in the results: **Section 5.1.2 paragraph on Results on nonlinear projection of causal variables**, “*This observation complements the identifiability result that recovery of latent variables is possible only upto a permutation in latent causal models [1,4] for general nonlinear mappings between causal variables and low-level data*”.
> - As per your suggestion, we have also **included a** **discussion on identifiability** in our revision in a new, separate section A.9 (”Discussion on identifiability”) of the appendix as well cited these works in the last paragraph of the related works section.
>
> [1] Brehmer, J., Haan, P. De, Lippe, P. & Cohen, T. Weakly supervised causal representation learning. in *ICLR2022 Workshop on the Elements of Reasoning: Objects, Structure and Causality* (2022).
>
> [2] Kartik Ahuja, Jason Hartford, & Yoshua Bengio. Weakly supervised representation learning with sparse perturbations.
>
> [3] Kartik Ahuja, Divyat Mahajan, Vasilis Syrgkanis, & Ioannis Mitliagkas. Towards efficient representation identification in supervised learning.
>
> [4] Yuhang Liu, Zhen Zhang, Dong Gong, Mingming Gong, Biwei Huang, Anton van den Hengel, Kun Zhang, and Javen Qinfeng Shi. Weight-variant latent causal models, 2022.

---

> ### Author Response · Authors · 2022-11-18
> **Response to reviewer jaMU (6/n)**
>
> > I am especially unhappy about the confusion between random variables and their realizations (e.g., the definition in Sec. 4.1, the paragraph below Eq. (7), in the paragraph "Evaluation Metrics", ...). This makes the paper harder to read, e.g. when determining whether p(Z | ..) is a conditional distribution/density or its evaluation, and is simply sloppy.
>
> These issues have been addressed in our latest revision. The notation has now been made consistent and we have clearly defined the random variables at the end of the paragraph in section 4.1. All the equations and derivations have also been made consistent with this notation so it should now be clearer as to when we refer to random variable/its realization or a conditional density/its evaluation.
>
> > Regarding originality, I miss the discussion of earlier, closely related work and do not agree with the claim that the authors are "the first to study this task of learning latent SCMs from low level observations..."
>
> Thanks for pointing this out. We have included the related work you had referred us to in your comment and we have not dedicated a paragraph to discuss these works (section 3 Related Work, paragraph 3). A more detailed comment discussing the setting of these works (which is different, discrete latent variables, does not operate on low-level observations etc.), and their assumptions has been discussed in comments (3/n and 4/n of this thread). Additionally, to be precise, we have now modified the text to say "the first to study this setting of Bayesian learning of latent SCMs from low level observations...".
>
> > From a methodological point of view the authors derive a Variational Inference objective (with issues; see above), and extend the causal discovery method from the recently proposed BCD Nets paper by estimating the observation generation conditional....
>
> We have addressed these concerns in our modified derivation in the appendix.
>
> > In Section 5.2 it is not clear whether the causal ordering is given or learned. The generalization demonstration in Fig. 7 is not really conclusive; a quantitative evaluation would be desirable (also comparing to baselines).
>
> - We have now made it clear both in main text and in image captions where the causal ordering is given and where it is learned.
> - Regarding fig. 7, we found it hard to evaluate it quantitatively on a metric. However, if you have a particular metric in mind, we would be glad to include it. Regarding comparison to baselines, as has been already addressed in previous comments ([here](https://openreview.net/forum?id=w2mDq-p9EEf&noteId=uNkugUjIvN)), we are aware of no other methods to the best of our knowledge that is capable of learning latent SCMs from low-level data and/or image generation from unseen interventional distributions. We could have adapted [1] for this purpose and used it as a baseline, but unfortunately the code has not been released so far.
>
> [1] Brehmer, J., Haan, P. De, Lippe, P. & Cohen, T. Weakly supervised causal representation learning. in ICLR2022 Workshop on the Elements of Reasoning: Objects, Structure and Causality (2022).
>
> > - An essential discussion of identifiability of the true latent SCM, and/or causal consistency of the learned SCM is missing; without it, making causal claims is contestable and the proposed method amounts to fitting a distribution over observed variables with a hierarchical latent space model.
> > - The submitted work lacks substantially in the discussion of and experimental comparison to related work.
> > - Issues of correctness/preciseness in the derived utility need to be addressed.
> > - The strong assumptions made should be concisely presented, compared to related work, and at least well justified.
>
> These concerns have been addressed as well:
> - Section A.9 is a discussion on identifiability
> - More discussion of related work has been added and we have cited all the related work you had mentioned as well (section 3, para 3)
> - This has been addressed in the equations and the derivation in section A.1
> - These have been addressed under a new section on Limitations (section 6)

---

> ### Author Response · Authors · 2022-11-24
> **Further questions to be addressed?**
>
> Dear Reviewer jaMU,
>
> Thank you again for your review. Please let us know if you have further questions so we can continue the discussion and see if there are any concerns left. If there are no further questions, we would be happy if you could consider raising the score.

---

> ### Author Response · Authors · 2022-12-07
> **Discussion period coming to an end; any remaining concerns?**
>
> Dear Reviewer jaMU,
>
> This is a gentle reminder that the discussion period is coming to an end. We have addressed all concerns brought up in our detailed responses and we would like to know if there are any remaining concerns? If there are no remaining concerns, adjusting the rating accordingly would be helpful.

---

> > ### Comment · Reviewer_jaMU · 2022-12-07
> > **identifiability**
> >
> > Thanks for the detailed response, and for the modifications to the paper (and sorry for the late reply).
> > While, I see that the paper has been improved in various aspects, my main concern remains the identifiability of the model. I agree with reviewer Aq59 that I am not convinced of theoretical identifiability. Although the focus of this paper is on the learning side, I still think that this is a crucial property for work on causal models.

---

> > > ### Author Response · Authors · 2022-12-09
> > > **On identifiability: Causal mechanisms are invertible, differentiable, and the inverse is differentiable as well**
> > >
> > > Thanks for pointing this out. It is true that the paper is not about theoretical identifiability since this is not the claim of the paper and has already been studied in other works. However, we have now addressed reviewer Aq59's concern which should also address the identifiability concern you are referring to. The main point of concern seems to be that it is hard to see how their identifiability results extend to our setting (because of differentiability and invertibility of causal mechanisms).
> > >
> > > Please refer page 3 of [Ref [1]](https://arxiv.org/abs/2203.16437) which states:
> > > > An SCM entails a unique solution s : $\varepsilon$ → $\mathcal{Z}$ defined by successively applying the causal mechanisms. We require the causal mechanisms to be pointwise diffeomorphic, that is, for any value of the [parents](https://arxiv.org/abs/2203.16437) $z_{pa_i}$ we have that $f_i(·; z_{pa_i})$ is invertible, differentiable, and its inverse is differentiable.
> > >
> > > The footnote at the end of the page includes the following statement which clarifies the concern:
> > > > Under some mild smoothness assumptions, any SCM can be brought into this form by elementwise redefinitions of the variables, preserving the observational and interventional distributions.
> > >
> > > Thus, we are still in the regime of [1] and the identifiability results extend. Further, to see this concretely for our linear Gaussian additive noise SCM case, consider $f_i(\varepsilon_i; z_{pa_i})$ which locally maps $\varepsilon_i \rightarrow \mathcal{Z}_i$ according to:
> > >
> > > $\sum\limits_{p \in {pa_i}} w_{ip} z_p + \varepsilon_i$
> > >
> > > - $f_i(\varepsilon_i; z_{pa_i})$ is a linear function and is differentiable with respect to $\varepsilon_i$
> > > - The inverse $g_i(.; z_{pa_i}) = f_i^{-1}(.; z_{pa_i})$ exists and maps $\mathcal{Z}_i \rightarrow \varepsilon_i$ according to:
> > >
> > > $\sum\limits_{p \in i \cup pa_i } \widetilde{w}_{ip} z_p$ where
> > >
> > > $\widetilde{w}_{ip} = 1$ if $p=i$ and
> > >
> > > equal to $-w_{ip}$ otherwise
> > > - The inverse $g_i(.; z_{pa_i}) = f_i^{-1}(.; z_{pa_i})$ is also a linear function and is differentiable with respect to $z_i$
> > >
> > > We hope this clarifies your question about identifiability. Please let us know if you have further questions and we would be glad to answer them. Given we posted many detailed responses (6 of them) apart from general comments, additional experiments (extending results to the non-equal noise variance, stochastic observation generation function, and improved results for learning permutation in linear projection setting) and modifications/additions to the text of the paper, it would helpful if you would consider raising the score accordingly.
> > >
> > > [1] Brehmer, J., Haan, P. De, Lippe, P. & Cohen, T. Weakly supervised causal representation learning. in ICLR2022 Workshop on the Elements of Reasoning: Objects, Structure and Causality (2022).

---

### Author Response · Authors · 2022-11-17
**General comments**

We thank the reviewers for their valuable comments and for their recommendations to improve our submission.

We are happy that the reviewers noticed the strengths of the work:
- "The authors do an excellent job of communicating the problem statement, the problem's technical challenges, and their solution. It involves a number of technical details to make the approach tractable. Overall the submission is very clearly written, is of high quality, and appears to be novel and reproducible" (pPfL)
- "The work is novel, in particular in respect to the learning of a latent causal model" (VekW)
- "The idea of a Bayesian approach to the latent structure learning problem is is certainly an interesting and promising one" (jaMU)
- "The authors estimate the latent causal model in a Bayesian way, which is different from previous estimation methods, as far as I know" (oCT2)
- "It makes some progress on the hard problem of learning the latent causal structures of interest from the ﻿unstructured dataset. The problem is challenging and the authors design a new method to address it" (Aq59)

Reviewers also brought up concerns, all of which we have addressed in our responses and/or revisions. Here is a summary of the main concerns of the reviewers and how we have addressed them:

**1. Discussion on identifiability**

Reviewers (jaMU, pPfL) pointed out that the paper did not discuss identifiability. We have **accordingly included a new section in the appendix (A.9) titled "Discussion on identifiability"**. Reviewers Aq59 and oCT2 point out that the paper lacks theoretical identifiability. We would like to clarify that this work is not about giving theoretical identifiability guarantees (though we provide strong empirical support for it in figures 4, 5, 6, 20, 21 in the paper). In fact, some papers already discuss identifiability in these settings which we have pointed to in our response to reviewers. As mentioned in these responses (see [here](https://openreview.net/forum?id=w2mDq-p9EEf&noteId=z-36XCFC1u), [here](https://openreview.net/forum?id=w2mDq-p9EEf&noteId=yWkkywi_S_) and [here](https://openreview.net/forum?id=w2mDq-p9EEf&noteId=ldrdoiudWZ)), identifiability does not imply learnability and our focus is an algorithm that learns latent SCMs from low-level data.

**2. Assumption related concerns**

Our experiments initially assumed a deterministic observation generating process (jaMU) and an equal noise variance (jaMU, Aq59) on our linear Gaussian latent SCM. Figure [4](https://ibb.co/0cPT6j3) and [5](https://ibb.co/SxsGvVH) were with these assumptions. We extended our experiments to settings without these assumptions, and summarize them in figure [20](https://ibb.co/YTgH25s) and [21](https://ibb.co/VBpmS1M)(also in the appendix). **Our proposed method continues to work even without these assumptions**. You can see the responses [here](https://openreview.net/forum?id=w2mDq-p9EEf&noteId=M9YxuVoAXB) and [here](https://openreview.net/forum?id=w2mDq-p9EEf&noteId=oArjpAhmM9F).

Some reviewers had concerns with other assumptions (some of which are common assumptions in the literature, for example, see [here](https://openreview.net/forum?id=w2mDq-p9EEf&noteId=oArjpAhmM9F)): we also addressed these in individual response to reviewers and acknowledged all assumptions under a new section 6 for clarity. Further, we also made a [tabular comparison](https://i.imgur.com/SdF1no0.png) of assumptions in related work and note that we make fewer/milder assumptions than these works (related discussions: [here](https://openreview.net/forum?id=w2mDq-p9EEf&noteId=uNkugUjIvN) and [here](https://openreview.net/forum?id=w2mDq-p9EEf&noteId=UFcSPc6tkzF))

**3. Concerns with ability of our model to infer SCM parameters**

Some reviewers (oCT2, VekW) brought up that in [figure 4](https://ibb.co/0cPT6j3), the MSE scores of the proposed method appears slightly worse than baselines in the case where the permutation is not given (plot in bottom row, last column). We noticed that **training for slightly longer fixes this issue** and it outperforms the baselines in this case as well (see [figure 20](https://ibb.co/YTgH25s)). A more detailed response can be found in the individual response to reviewers.
**Related discussions**: [our answer to concern #6 of VekW](https://openreview.net/forum?id=w2mDq-p9EEf&noteId=CQ-wHJVVB9) and [response to concern #2 of oCT2](https://openreview.net/forum?id=w2mDq-p9EEf&noteId=W1K3bIPHMls).

---

> ### Author Response · Authors · 2022-11-17
> **General comments (contd)**
>
> **4. More comparison with baselines**
>
> Reviewers (jaMU, oCT2) pointed out that comparison with more baselines (point-estimate or Bayesian approaches) would be beneficial. As mentioned in section 5 regarding baselines (paragraph 2) there are no other methods to our knowledge that solves this problem of learning a latent SCM from low-level data (high dimensional vectors or pixels). There are a few works in the space of latent confounder settings or in cases where some variables of the SCM are observed and others are not (papers [1-6] in jaMU's review) -- but this is a very different setting to ours. We have also given how and why this is a different setting (see [this response](https://openreview.net/forum?id=w2mDq-p9EEf&noteId=uNkugUjIvN) and this [table](https://i.imgur.com/SdF1no0.png)). The only work that is in our setting and could be a potential baseline is paper [7] given in jaMU's review. However, after correspondence with the authors we found that the code is not available yet and we would be happy to add them as a baseline after the code is released (see [this response](https://openreview.net/forum?id=w2mDq-p9EEf&noteId=gztR1Gq1be)). Our [response to concern #3 of oCT2](https://openreview.net/forum?id=w2mDq-p9EEf&noteId=XJEmrgEWHs) is also relevant here.
>
> We are happy to address any further questions to alleviate the reviewers' concerns and thank the reviewers for their valuable comments.

---

### Decision · Program_Chairs · 2023-01-20

**Decision:**

Reject

**Justification For Why Not Higher Score:**

Concerns regarding identifiability of the underlying model as needed for causal claims

**Justification For Why Not Lower Score:**

N/A

**Metareview: Summary, Strengths And Weaknesses:**

This paper considers the problem of learning SCMs with latent variables in the setting of "causal representation learning". The authors propose a Bayesian approach assuming the number of latent variables, their causal order, and the intervention targets are known. Most seriously, reviewers raised concerns regarding identifiability of the underlying model, which is a prerequisite for causal claims. During discussion, the authors appealed to some prior work to address this, however, the connection with the current work is not clear. A major revision is needed to prove this rigourously and make the connection with prior work clear.

**Summary Of Ac-Reviewer Meeting:**

N/A